# Extracellular LGALS3BP regulates neural progenitor position and relates to human cortical complexity

Christina Kyrousi[1,14], Adam C. O'Neill[2], Agnieszka Brazovskaja[3], Zhisong He [3,4], Pavel Kielkowski[5,15], Laure Coquand[6], Rossella Di Giaimo [1,7], Pierpaolo D' Andrea[1], Alexander Belka [1], Andrea Forero Echeverry[1], Davide Mei[8], Matteo Lenge [8], Cristiana Cruceanu[9], Isabel Y. Buchsbaum[1,10], Shahryar Khattak [11,16], Guimiot Fabien[12], Elisabeth Binder [9], Frances Elmslie[13], Renzo Guerrini[8], Alexandre D. Baffet [6], Stephan A. Sieber [5], Barbara Treutlein [3,4], Stephen P. Robertson [2✉] & Silvia Cappello [1✉]

Basal progenitors (BPs), including intermediate progenitors and basal radial glia, are generated from apical radial glia and are enriched in gyrencephalic species like humans, contributing to neuronal expansion. Shortly after generation, BPs delaminate towards the subventricular zone, where they further proliferate before differentiation. Gene expression alterations involved in BP delamination and function in humans are poorly understood. Here, we study the role of LGALS3BP, so far known as a cancer biomarker, which is a secreted protein enriched in human neural progenitors (NPCs). We show that individuals with *LGALS3BP* de novo variants exhibit altered local gyrification, sulcal depth, surface area and thickness in their cortex. Additionally, using cerebral organoids, human fetal tissues and mice, we show that *LGALS3BP* regulates the position of NPCs. Single-cell RNA-sequencing and proteomics reveal that LGALS3BP-mediated mechanisms involve the extracellular matrix in NPCs' anchoring and migration within the human brain. We propose that its temporal expression influences NPCs' delamination, corticogenesis and gyrification extrinsically.

---

[1] Max Planck Institute of Psychiatry, 80804 Munich, Germany. [2] Department of Women's and Children's Health, University of Otago, 9054 Dunedin, New Zealand. [3] Max Planck Institute for Evolutionary Anthropology, 04103 Leipzig, Germany. [4] ETH Zurich, Department of Biosystems Science and Engineering, 4058 Basel, Switzerland. [5] Department of Chemistry, Chair of Organic Chemistry II, Center for Integrated Protein Science (CIPSM), Technische Universität München, Garching, Germany. [6] Institut Curie, PSL Research University, CNRS, UMR 144, 26 rue d'Ulm, F-75005 Paris, France. [7] Department of Biology, University of Naples Federico II, 80126 Naples, Italy. [8] Neuroscience Department, Children's Hospital A. Meyer-University of Florence, 50139 Florence, Italy. [9] Department of Translational Research in Psychiatry, Max Planck Institute of Psychiatry, 80804 Munich, Germany. [10] Graduate School of Systemic Neurosciences, Ludwig-Maximilians-University, 82152 Munich Planegg, Germany. [11] DFG-Research Center and Cluster of Excellence for Regenerative Therapies (CRTD), School of Medicine, Technical University Dresden, 01307 Dresden, Germany. [12] Unité de Foetopathologie, Assistance Publique-Hôpitaux de Paris, CHU Robert Debré, F-75019 Paris, France. [13] South West Thames Regional Genetics Service, St George's, University of London, London SW17 0RE, UK. [14] Present address: First Department of Psychiatry, Medical School, National and Kapodistrian University of Athens, Greece and University Mental Health, Neurosciences and Precision Medicine Research Institute "Costas Stefanis", Athens, Greece. [15] Present address: Department Chemie Ludwig-Maximilians-Universität München Butenandtstr. 5-13, 81377 München, Germany. [16] Present address: Royal College of Surgeons Ireland (RCSI) in Bahrain, Adliya, Kingdom of Bahrain. ✉email: stephen.robertson@otago.ac.nz; silvia_cappello@psych.mpg.de

Cortical development is a tightly regulated process in which neural progenitor cells (NPCs) in the ventricular (VZ) and subventricular zones (SVZ) generate cortical neurons[1]. Although the basic molecular and cellular mechanisms governing corticogenesis in mammals are shared, the substantial differences regarding the shape and size of the cortex between primates (and in particular humans) and other mammals, suggest that specific changes have occurred during evolution[2–5]. Dysregulation of these mechanisms as a consequence of genetic alterations causes cortical malformations in humans, such as microcephaly, lissencephaly, and periventricular heterotopia (PH)[6–11].

One possible explanation for the massive increase in cortical size in primates was proposed to be the expansion of basal progenitors (BPs), namely intermediate progenitors (IPs) and basal radial glial cells (bRGs). Generated from apical radial glial cells (aRGs) in the VZ, both types of BPs delaminate to the inner and outer SVZ (iSVZ and oSVZ, respectively). IPs perform mainly symmetric neurogenic divisions but in humans, it was shown that they also have a proliferating capacity[12,13], while bRGs undergo several rounds of proliferating divisions before terminal differentiation. bRGs are found in higher numbers in gyrencephalic compared to lissencephalic species and thus, have been proposed to be central for the generation of gyri and sulci of the cerebral cortex[4]. Thereby, both BPs contribute to the increased number of neurons found in the human cortex[14–16].

Significant differences between the transcriptome of NPCs in the germinal zones generating gyri compared to those generating sulci have been demonstrated[17], and consequently it was proposed that certain genes, which are newly evolved or whose expression and/or function are altered in gyrencephalic species, can control the delamination and function of BPs. Such factors would also be hypothesized to mediate the expansion of the cerebral cortex and hence contribute to gyrification. Numerous genes have already been implicated in the generation of BPs facilitating gyrification[15,18–22] with recent insight also outlining a role for mechanical forces[23]. The extracellular matrix (ECM) has also been implicated in both neocortical expansion and cortical folding[18,24–26]. However, the nature of the mechanistic connection between the molecular, cellular, and mechanical factors influencing the generation and function of BPs in gyrification remains largely unknown[27].

To this end, we studied the role of galectin-3 binding protein (LGALS3BP), a secreted protein that interacts with several members of the ECM such as integrins, fibronectins, galectins, laminins, and tetraspanins[28–31], and has been well studied in cancer biology. Recently, it was shown that the *LGALS3BP* mRNA, although almost undetectable during mouse corticogenesis[32], is enriched in human NPCs[18,24] and bRGs[33–35]. Here we show that LGALS3BP is enriched in human NPCs and its function is of great importance in human corticogenesis since changes in local gyrification are shown in individuals with de novo genetic variants in *LGALS3BP*. We propose that LGALS3BP is secreted via extracellular vesicles (EVs) and its interaction with the ECM, collagens, and tetraspanins regulates apical anchoring of aRGs, leading to BP delamination, and consequently cortical folding in humans.

## Results

**Expression of *LGALS3BP* in human NPCs**. Recently, it was proposed that the niche where NPCs and neurons are generated and function is essential for proper cortical development, and that mechanical forces, as well as specific extracellular environment, may contribute to human cortical development[25,36]. However, little is known so far on the human-specific mechanisms regulating corticogenesis. This lack of information may also be because the most commonly used model for corticogenesis is the mouse, a lissencephalic species, and thus it becomes apparent that the use of other models is necessary. Towards identifying human-specific niche regulators and human-specific molecular and cellular mechanisms regulating human NPC generation and differentiation, we hypothesized that genes that are highly expressed in the developing human cortex (where BPs are expanded) but are with a reduced expression in mice (where BPs are restricted), would be prominent regulators of such processes. Thus, we focused on the gene *LGALS3BP*, whose mRNA according to previous work is expressed at high levels in aRGs, IPs, and bRGs in humans, is downregulated in human neurons (Supplementary Fig. 1a) and concurrently is almost undetectable in the developing mouse brain[18,24,32]. Additionally, *LGALS3BP* mRNA expression in the developing ferret brain is twofold higher in the neurogenic regions that form gyri, compared to those generating sulci[17], and its expression is increased during the evolution of the brain from macaque to chimpanzee and humans[37].

To gain insight into the temporal and spatial localization of *LGALS3BP*, we generated human induced pluripotent stem cell (iPSC)-derived cerebral organoids (COs)[38] (Supplementary Fig. 1b) and examined *LGALS3BP* expression longitudinally over COs development. *LGALS3BP* mRNA has its peak of expression shortly after the known aRG marker *PAX6* and before BP markers like *EOMES* and *HOPX* (Fig. 1a), a pattern which is also observed in the human fetal brain as shown from the BrainSpan dataset (https://www.brainspan.org/). To further investigate the *LGALS3BP* gene expression at the single-cell level in COs, we performed single-cell RNA-sequencing (scRNA-seq) in 60 days' (60d) COs and showed that *LGALS3BP* is enriched in progenitor cells but is progressively reduced upon neuronal differentiation (Fig. 1b and Supplementary Data 1). These data were confirmed by qPCRs performed in iPSC-derived NPCs and neurons in 2D (Supplementary Fig. 1c). In addition, we performed in situ hybridization in COs at 30d, 50d, and 60d and showed that *LGALS3BP* is expressed in a salt and pepper fashion in NPCs, suggesting its upregulation in a specific subpopulation of progenitors (Fig. 1c–e and Supplementary Fig. 1d–i').

LGALS3BP was shown to be a component of the ECM in cancer tissues, thus, to investigate whether LGALS3BP is expressed in COs at the protein level and whether it is secreted from NPCs and neurons, we have analyzed the full proteome (full lysate) and secretome (culture medium) of control COs at 60d, a developmental stage where mRNA of *LGALS3BP* is highly present. This analysis confirmed the protein expression and secretion from human cortical cells (Supplementary Data 2 and 3). Additionally, immunofluorescence performed in mice, COs, or in human fetal cortical tissues revealed specific dotty staining of LGALS3BP (highlighted with arrows) in humans but not in mice (Fig. 1f–q and Supplementary Fig. 1j–n), which is in line with its secretion and presence in vesicles as recently reported[39–41], and that its expression is mainly in the neurogenic zones of the human developing cortex, namely the VZ, iSVZ, and oSVZ. Moreover, the LGALS3BP+ punctae are in close correlation with the progenitor marker SOX2 and not with the neuronal marker NEUN (Fig. 1o–q and Supplementary Fig. 1g–i). Collectively, these data suggest that *LGALS3BP* has a very dynamic expression and localization both at the cellular and temporal level during human cortical development. We, thus, hypothesized that it could shape the extracellular niche of human NPCs, contributing to the dynamics of human NPCs' transition and influence the process of gyrification of the human cerebral cortex.

**Individuals with *LGALS3BP* variations have cortical changes**. Expression of *LGALS3BP* is enriched in the developing brain in

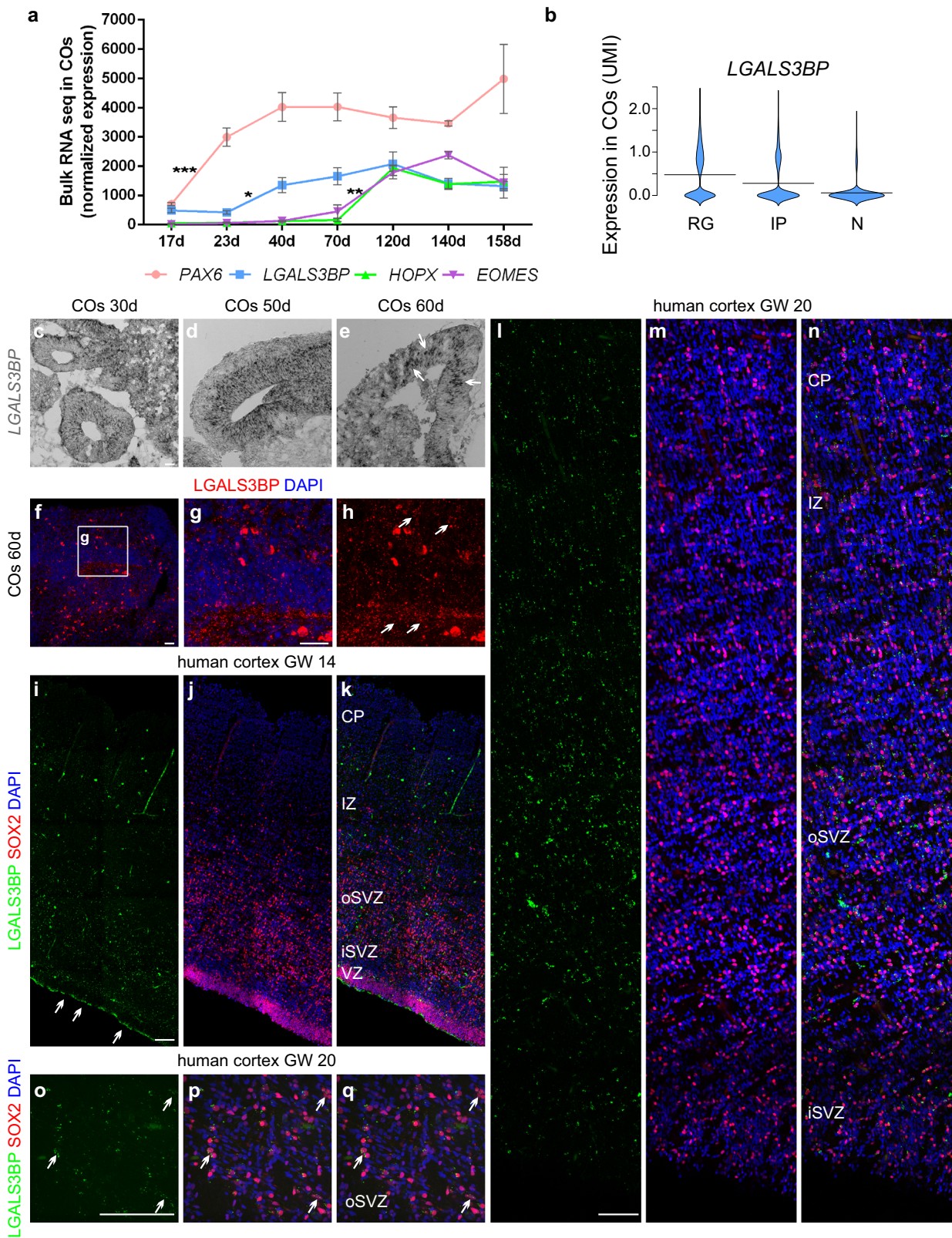

humans and assessment of data from control individuals (derived from the NHLBI-ESP6500 dataset) demonstrate that the coding region of *LGALS3BP* is depleted of genetic variation in human populations with metrics placing it within the top 10% of genes intolerant to non-conservative changes with a specific Residual Variance Intolerance Score (RVIS) of 8.9%[42]. Therefore, we hypothesized that variants within *LGALS3BP* could result in

defective neurogenesis in humans. The enrichment of loci with rare de novo variants in individuals with the neurodevelopmental condition called periventricular nodular heterotopia (PH) was recently described, suggesting the involvement of these genes in the pathogenesis of this disease[11,43]. In this study[11], one individual with a de novo missense variant in *LGALS3BP* (individual 1, c.1108 G > A [p.Glu370Lys]; RefSeq NM_005567.4) was

**Fig. 1 LGALS3BP expression in human cortical progenitors. a** Bulk RNA-sequencing data performed in COs at different stages. LGALS3BP has a peak of expression after apical progenitor marker PAX6 and prior to basal progenitor markers EOMES and HOPX. Data are shown as mean ± SEM, batches = 3, organoids = 3. Statistical analysis was performed using two-way ANOVA followed by Tukey's multiple comparison test, *$p < 0.05$, **$p < 0.01$, ****$p < 0.0001$. In (**a**), $p = 0.0026$ in ctrlvsY366Lfs. **b** Violin plots from scRNA-seq data performed in 60d control COs showing LGALS3BP mRNA expression levels in RGs, IPs, and N, batches = 1, organoids = 2, cells = 9290. **c–e** In situ hybridization depicting mRNA expression of LGALS3BP in 30d, 50d, and 60d control COs. Arrows show basally located LGALS3B+ cells, batches = 1, organoids = 3, ventricles = 3. **f–h** Micrograph of control CO sections immunostained with specific LGALS3BP antibody. **i–q** Micrograph of human fetal cortical sections immunostained as depicted in the panels at GW14 (**i–k**) and 20 (**l–q**). Arrows show LGALS3BP+ cells. Scale bar: 30 μm in (**c–h**), 500 μm in (**i–n**), and 100 μm in (**o–q**). See also Supplementary Fig. 1. Abbreviations: d: days, RG: radial glial cells, IP: intermediate progenitors, N: neurons, GW: gestational week, VZ: ventricular zone, iSVZ: inner subventricular zone, oSVZ: outer subventricular zone, IZ: intermediate zone, CP: cortical plate.

diagnosed with developmental delay, autism, dysarthria, ataxia, and focal seizures at 3 years of age. His MRI demonstrated PH in the right temporal horn and within the contralateral peri-hippocampal region, close to a rounded and under-rotated left hippocampus in addition to cerebellar vermian dysgenesis with an accompanying mega cisterna magna (Supplementary Fig. 2a, b–d). Two further individuals with de novo missense variants in LGALS3BP (individual 2, c.880 C > T [p.Glu294Lys]; RefSeq NM_005567.4; and individual 3, c.1580 T > C [p.Glu527Gly]; RefSeq NM_005567.4) were identified within the Deciphering Developmental Disorders (DDD) study ($n = 1133$)[44] (Supplementary Fig. 2a). Phenotypically, individual 2 was diagnosed with prenatal onset microcephaly (occipitofrontal head circumference −5.5 SD) and global neurodevelopmental delay. An MRI demonstrated a primitive sulcal pattern and a lack of folding of the operculum. There were no seizures or radiological evidence for PH. No clinical data could be ascertained for individual 3. All three de novo variants in these individuals lie within the most intolerant sub-region of LGALS3BP[45]—namely the exon 5 (Supplementary Fig. 2e)—suggesting that they may be functionally disruptive, and thereby contributing causally to the neurodevelopmental phenotypes observed. Although individuals 1 and 2 were described with different developmental diseases, by analyzing the morphology of their brains via brain MRI imaging with age-matched morphometric analysis[46,47], we noticed similar defects in the formation of their brains. In particular, cluster analysis in both individuals revealed changes in cortical thickness, local gyrification index (LGI), surface area, and sulcal depth (Fig. 2a, a'), which, even though were not perfectly overlapping in the brain regions of the two individuals, suggest that overall LGALS3BP variants affect cortical formation.

To gain insight into the mechanisms which were affected in the clinically observed LGALS3BP genetic variants, by using CRISPR/Cas9 genome editing we generated iPSC lines carrying either the variant found in individual 1 (p.E370K, from now on referred to as E370K) in heterozygosity to mimic exactly the situation observed in individual 1 or a two base-pair deletion before the position of the variant in this individual, which resulted in a frameshift (p.Y366Lfs*443, from now on referred to as Y366Lfs) in homozygosity to avoid a potential mask of phenotype due to the existence of a functional allele in the case of the heterozygous E370K (Supplementary Fig. 2f and Supplementary Data 4). We prioritized individual 1 because individual 2 was only 1 year old when imaged which, due to incomplete myelination, makes it difficult to properly assess the cortical thickness, LGI, surface area, and sulcal depth. We generated COs derived from isogenic control and from the genetically edited iPSCs, E370K, and Y366Lfs, which show reduced LGALS3BP expression compared to control (Supplementary Fig. 2g–j). We then monitored their development at different stages. During the early stages of development, genetically edited COs exhibited a delay in their growth as indicated by their smaller perimeter which, however, was normalized within a few days (Supplementary Fig. 2k–q).

Also, no significant changes were observed either in the total number of progenitors, PAX6+, SOX2+, and HOPX+ populations (Fig. 2b, c and Supplementary Fig. 2r–t), or in the total number of neurons, DCX+ and NEUN+ populations (Supplementary Fig. 2u–ab'). However, we observed differences in the distribution of NPCs. A higher number of NPCs performed mitosis at the apical surface in proximity to the ventricles compared to controls (Fig. 2d–h). Interestingly, we noticed that both general RG markers, such as FABP7, and bRG markers, such as HOPX, had different distributions in the mutant COs compared to controls. Namely, we observed increased number of FABP7+ cells located within the VZ (Fig. 2i–l) and an increased number of ventricles containing more than 10 HOPX+ cells localized apically and demonstrating an apical accumulation in the genetically edited COs (almost 70% and 75% in the E370K and the Y366Lfs mutant COs, respectively, compared to 30% in control COs) (Fig. 2m–p). This suggests that a higher number of NPCs, either apical or basal, may divide at apical positions within the mutant backgrounds.

Since individual 1 also exhibits PH, we explored the idea that PH could be caused by the misplacement of NPCs. To test this hypothesis, we assessed the neuronal populations within the mutant COs and found that, although their total number was unaffected, there was a substantial number of neurons positioned in the VZ (Supplementary Fig. 2u–w', z–ab'). We, therefore, quantified the percentage of ventricles that featured ectopic neurons in the VZ and found that almost 60% and 75% of the E370K and Y366Lfs mutant ventricles, respectively, had neurons within the progenitor zone in contrast to 28% in the isogenic control (Supplementary Fig. 2ac). These percentages were very similar to the previously quantified ventricles with non-delaminated (apical) HOPX+ BPs (Fig. 2p) and could explain the morphological changes observed in the individuals with LGALS3BP variants. Apart from the neuronal misposition, we did not observe any other cortical defects even though both individuals 1 and 2 exhibit changes in the LGI, sulcal depth, and cortical surface. This is not surprising since gyrification cannot be assessed using COs since there is no normal folding in the cortical plate of COs—a limitation of this model system. Nevertheless, many of the changes observed in the two individuals are located in the deep areas of the MRIs which could reflect the ectopic neurons. All the above suggest that the different mutations in the exon 5 of the LGALS3BP gene or its complete loss impact its proper expression and results in changes in the development of the neuroepithelium in the human cortex.

To determine whether LGALS3BP is sufficient to influence human neurogenesis by changing their number or by misplacing NPCs and neurons during development, we manipulated its expression in control COs and organotypic slices of the human fetal brain (Supplementary Fig. 3a, a'). Overexpression of LGALS3BP via electroporation in COs at 40d was chosen for further analysis because this is a stage before the peak of LGALS3BP expression and before the significant increase of the

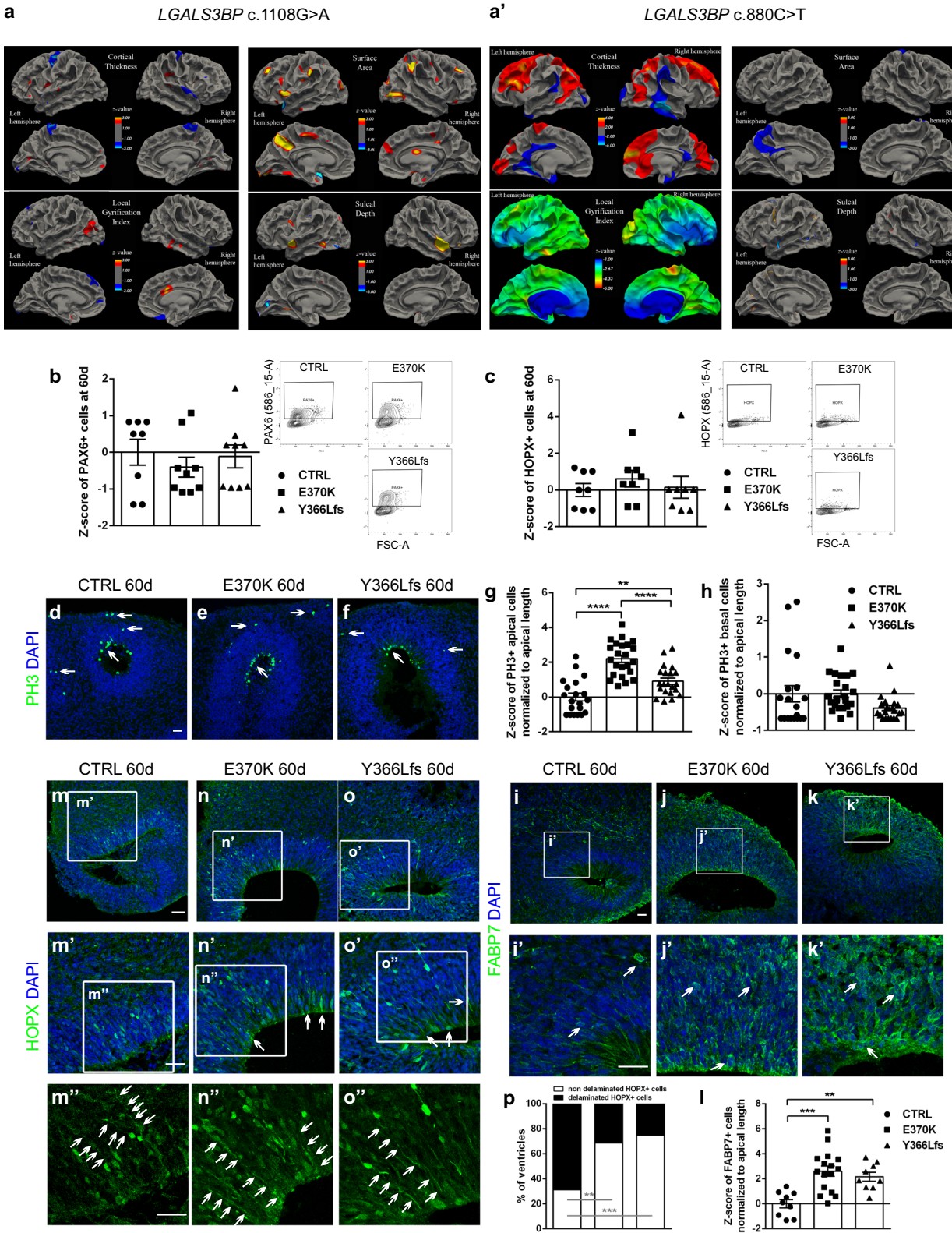

BPs (*EOMES+* IPs and *HOPX+* bRGs) in the CO cultures (as shown in Fig. 1a). Analysis at 4 days later (4dpe, days post electroporation) showed that *LGALS3BP* overexpression changed the morphology of NPCs from a radial-bipolar to a more multipolar shape and led to the loss of their apical attachment (Supplementary Fig. 3b, c). Also, overexpression of *LGALS3BP* in COs at 40d resulted in a reduced number of SOX2+ apical NPCs

in the VZ (Supplementary Fig. 3d–g', p), increased number of HOPX+ BPs (Supplementary Fig. 3h–k', q, r), and increased neuronal differentiation suggesting a premature differentiation in these COs (Supplementary Fig. 3l–o', s). Next, to further assess the function of *LGALS3BP* in human NPCs, we overexpressed *LGALS3BP* in human fetal brain sections obtained from gestational week (GW) 16 fetuses and analyzed the number of

**Fig. 2 Genetic Variants within *LGALS3BP* in humans associated with cortical malformations. a, a′** Morphometric analysis of the MRIs of individual 1 (**a**) and 2 (**a′**) showing changes in cortical thickness, local gyrification index, surface area and sulcal depth in different positions of the cortex. Statistical analysis of morphometric data was performed by a two-tailed paired *t*-test which compares each individual with a template obtained by averaging 8 age-matched controls. Cortical maps were clustered and corrected for multiple comparisons. **b, c** FACS plots and respective quantification depicting the sorting of PAX6+ (**b**) or HOPX+ (**c**) cells in 60d old control, E370K, and Y366Lfs mutant COs. The total proportion of the apical and basal progenitor cells is not changed in mutant COs. Data are shown as *Z*-scores ±SEM, batches = 2, organoids = 9. **d–f, i–k′, m–o″** Micrographs of sections of 60d control, E370K, and Y366Lfs mutant COs immunostained for PH3 (**d–f**), FABP7 (**i–k′**) or HOPX (**m–o″**). Arrows indicate the PH3+, FAPB7+, or HOPX+ cells which retain the apical contact with the ventricular zone. The apical and basal processes of the HOPX+ cells as indicated by the staining (arrows along the length of the processes) show the cells which should typically lose their apical processes, they tend to retain them and keep the attachment to the apical membrane in the mutant COs. **g, h, l** Quantification of the apically (**g**) or basally (**h**) located PH3+ cells and the number of FABP7+ cells within the VZ normalized to the apical membrane length (**l**) are shown as *Z*-scores ±SEM. Statistical significance was based on two-sided Mann-Whitney *U* test ** *p* < 0.01, *** *p* < 0.001, **** *p* < 0.0001, batches = 2, organoids = 6, ventricles = 20. In (**g**), *p* = 0.0026 in ctrlvsY366Lfs. In (**l**), *p* = 0.0005 in ctrlvsE370K, *p* = 0.0012 in ctrlvsY366Lfs. **p** Quantification of the percentage of ventricles having non-delaminated HOPX+ cells. Statistical analysis was based on one-tailed exact binomial test ** *p* < 0.01, *** *p* < 0.001, batches = 2, organoids = 6, ventricles = 15. In (**p**), *p* = 0.0023 in ctrlvsE370K, *p* = 0.0004 in ctrlvsY366Lfs. Scale bar: 30 µm. See also Supplementary Fig. 2. Abbreviations: CTRL: control, d: days.

SOX2+ cells in the oSVZ 7dpe for assessing mainly the number of bRGs[48] (Supplementary Fig. 3t–z). Similar to what was observed in COs, *LGALS3BP* overexpression resulted in an increased number of oSVZ SOX2+ NPCs—basally located NPCs. All the above pointed out that dysregulation of *LGALS3BP* either by mutation or by overexpression influences the position of human NPCs in the different neurogenic zones, the VZ and SVZ.

One of the mechanisms that promote the generation of BPs from aRGs and their translocation to the SVZ is the detachment from the apical side and the delamination. Since mutation or overexpression of *LGALS3BP* affects the position of NPCs, we examined whether LGALS3BP has a role in the apical anchoring and we analyzed the apical belt in COs. Immunostaining for β-catenin and F-actin (phalloidin immunostaining) was performed after *LGALS3BP* overexpression in COs at 4dpe (Fig. 3a–h). At the site of electroporation, but not in adjacent regions or control electroporation, β-catenin and F-actin expression were dramatically reduced (green and red arrowheads, respectively, in Fig. 3a–h) in almost 70% of the ventricles in the COs analyzed (Fig. 3i). Interestingly, in some cases, cells were submerged to the ventricular space (Fig. 3c–d) possibly due to the loss of the apical membrane integrity. In contrast, the thickness of the apical belt in both E370K and Y366Lfs genetically edited COs was strongly increased compared to controls as indicated by several apical junction markers, such as F-actin (Fig. 3j–l′), β-catenin (Fig. 3m–o), PALS1 (Fig. 3p–r), pan-cadherin (Fig. 3s–u), and the cilia protein ARL13B, as cilia are localized apically in RGs and therefore is an indirect indicator of the apical endfeet of aRGs (Fig. 3v–x). Quantification of the thickness of the apical belt based on some of its major components, namely the cytoskeleton (F-actin), members of the apical polarity complex (PALS1), and members of the adhering components (pan-cadherin), showed that the apical belt in mutant COs is almost three times thicker than in control COs (Fig. 3y–aa). Given that LGALS3BP was shown to cause centriolar accumulation[49], we also tested cilia dysregulation in mutant COs. Despite the increase in the total number of cilia (Fig. 3a, b), which can be explained by the increased number of progenitors found in the VZ of the mutant COs as it was previously described, we also observed a reduction in their length (Fig. 3a, c). We did not observe any further dysregulation of cilia like disorientation or increased number of cilia per cell suggesting no major malfunction of cilia. Taken together, forced local expression of *LGALS3BP* in aRGs resulted in disorganization of the apical belt, which we propose as the cellular mechanism leading to increased delamination of NPCs. In contrast, *LGALS3BP* mutations resulted in the thickening of the apical belt and the prevention of delamination of NPCs. This effect of LGALS3BP on the apical anchoring may also explain the

indirect mislocalization of neurons observed in mutant COs and individuals with *LGALS3BP* genetic variants.

**Mutant COs reveals cell non-autonomous function**. To decipher the molecular mechanisms through which LGALS3BP modulates human neurogenesis, NPC distribution, and gyrification, we firstly performed a whole-proteome analysis of the lysate isolated from the control and genetically edited COs in order to discriminate any changes in the protein levels between the three conditions—control, E370K, and Y366Lfs COs (Fig. 4a and Supplementary Data 2). We chose to perform this analysis in COs at 60d which is a time point when LGALS3BP expression shows a significant increase, while there is no significant increase of the BP markers (Fig. 1a). First, we validated that the levels of LGALS3BP in the mutant COs are reduced (Fig. 4a), and that the LGALS3BP levels are less reduced in the E370K COs (heterozygous line) than in the Y366Lfs COs (homozygous line). Then, the proteomic profile of the E370K and Y366Lfs COs was assessed by mass spectrometry. No significant changes between E370K and Y366Lfs COs were observed (Fig. 4a and Supplementary Data 2) and for that reason, we further analyzed the data combining the proteome from the two different mutant COs. GO Term analysis of the upregulated and downregulated proteins identified, revealed that the biological processes that were mainly dysregulated were those related to neurogenesis, neuron differentiation, and neuron development (Fig. 4b), which comes in line with the changes in cortical thickness and LGI in the 2 individuals with *LGALS3BP* genetic variants found and the phenotype observed upon manipulation of LGALS3BP in COs. Another process that was found highly dysregulated from this analysis was related to cellular component organization, cellular component biogenesis, and cellular component morphogenesis (Fig. 4b). Interestingly, the three highest cellular components that were found dysregulated were the extracellular organelle, extracellular vesicles, and extracellular exosomes, speaking for changes in the extracellular environment of these COs (Fig. 4b). For that reason, we then focused on the protein differences in the extracellular environment of the genetically edited COs by performing secretome analysis (mass spectrometry analysis performed on the proteins secreted in the culture medium; Fig. 4c and Supplementary Data 3). First, we observed that the two mutant forms of LGALS3BP were very little secreted because of their reduced initial expression levels and/or alterations in secretion. Second, similar to what was observed in the proteome, the two mutants did not differ significantly from each other and were further processed together (Fig. 4c). We then examined the most significant biological and cellular component processes that were dysregulated in the mutant COs and, in line with the proteome,

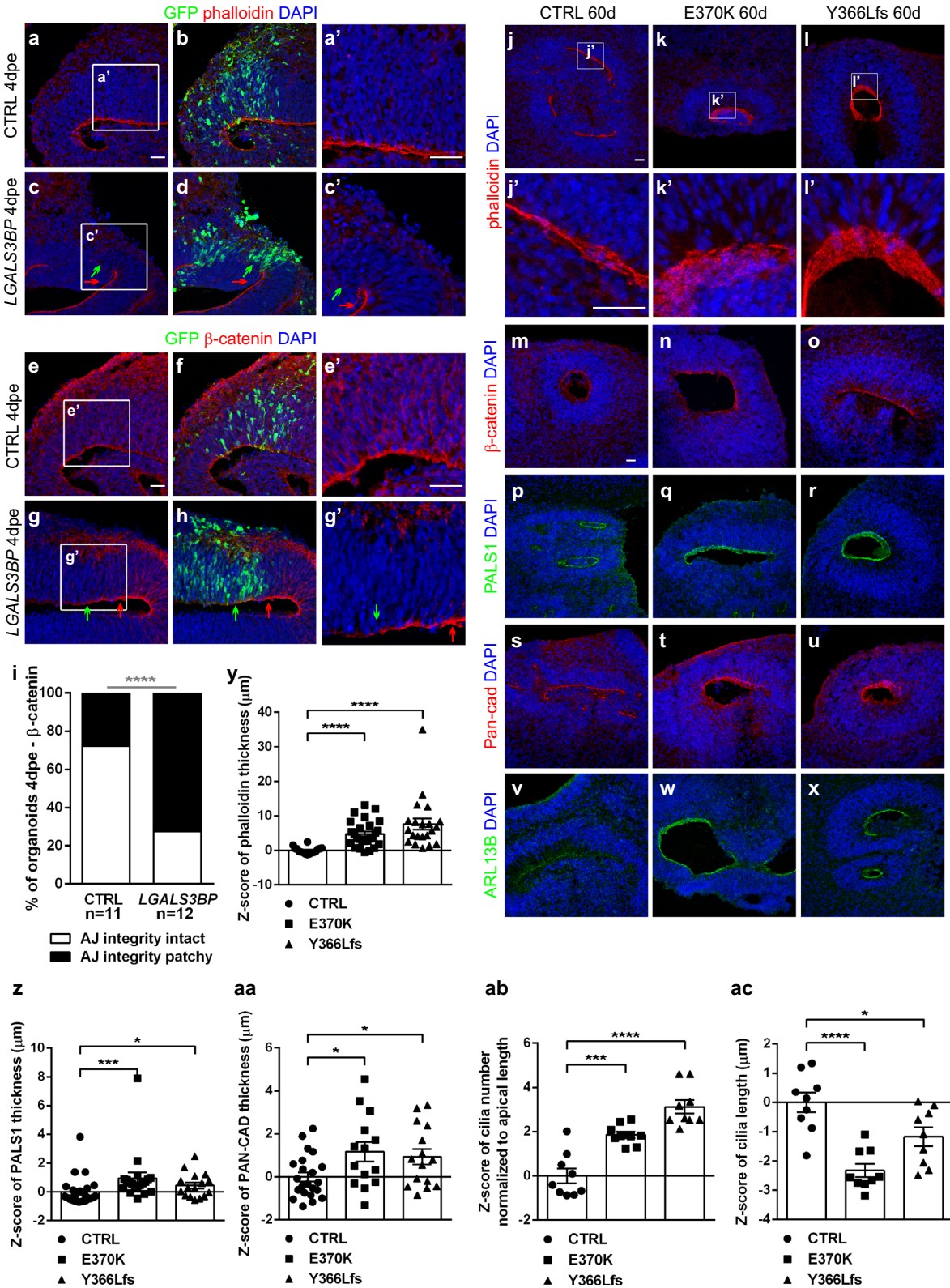

we observed that the extracellular structure organization, ECM, extracellular vesicles, and vesicle lumen turned up. All the above are of great importance taking also into account that LGALS3BP is secreted via vesicles in the size of exosomes and shows that the extracellular environment is influenced by the expression/secretion of LGALS3BP, indicating its cell non-autonomous role in human brain development.

To dissect the specific cell type(s) affected in the COs, we performed scRNA-seq analysis in control, E370K, and Y366Lfs COs. SPRING and clustering analysis of the RSS[37] integrated scRNA-seq-data-identified cells from all different areas in the developing brain (Fig. 4e–g, Supplementary Fig. 4a–d, and Supplementary Data 1), as expected when using intrinsic protocols for the generation of COs[50]. The dorsal telencephalic

**Fig. 3 LGALS3BP controls apical anchoring of human progenitor cells. a–h** Micrographs of sections of control COs immunostained as indicated in the panels after electroporation of control or *LGALS3BP* at day 40 and analyzed 4dpe and immunostained as indicated in the panels (phalloidin in (**a–d**), β-catenin in (**e–h**)). Green arrows depict the electroporated area and red arrows the adjacent area. White boxes indicate the area which is zoomed-in in corresponding pictures. **j–x** Micrographs of sections of day 60 control, E370K, and Y366Lfs mutant COs immunostained as indicated in the panels (phalloidin in (**j–l'**), β-catenin in (**m–o**), Pals1 in (**p–r**), Pan-cad in (**s–u**), Arl13b in (**v–x**)). White boxes indicate the area which is zoomed in in corresponding pictures. **i** Quantification of the percentage of COs with intact or patchy apical junction upon overexpression of *LGALS3BP* depicted in (**a–g'**). Data were statistically analyzed with one-tailed exact binomial test, ****$p < 0.0001$, batches = 2, organoids = 6, ventricles = 12. **y–ac** Quantification of the phalloidin (**y**), PALS1 (**z**) or PAN-CAD (**aa**) thickness depicted in (**j–u**) and cilia number (**ab**) and length (**ac**) depicted in (**v–x**). Data in (**z–ac**) are shown as $Z$-scores ±SEM, statistical significance was based on the two-tailed Mann-Whitney $U$ test *$p < 0.05$, ***$p < 0.001$, ****$p < 0.0001$, batches = 2, organoids = 6, ventricles=12. In (**z**), $p = 0.0003$ in ctrlvsE370K, $p = 0.0184$ in ctrlvsY366Lfs. In (**aa**), $p = 0.0271$ in ctrlvsE370K, $p = 0.0302$ in ctrlvsY366Lfs. In (**ac**), $p = 0.0012$ in ctrlvsE370K. Scale bar: 30 µm. See also Supplementary Fig. 3. Abbreviations: CTRL: control, dpe: days post electroporation, d: days, PAN-CAD: PAN-CADHERIN.

cells were aligned on a developmental pseudotime, which revealed the progression from cortical RGs via IPs to neurons (Fig. 4f and Supplementary Fig. 4c, d). This differentiation trajectory was preserved in all three genotypes (control, E370K, and Y366Lfs). Together, this suggests that basic developmental processes were maintained in *LGALS3BP* mutant COs. However, when comparing the proportion of each cell type in control and mutant COs (Fig. 4h and Supplementary Fig. 4b, e), we noticed that (i) the majority of the mutant CO cells were mainly with dorsal identity and (ii) the bRG population was strikingly less represented in the mutant COs (135 bRGs in control, 4 in Y366Lfs, and 15 in E370K), while the other cortical cell states were represented in similar numbers (Fig. 4h).

Next, we performed differential expression (DE) analysis in cortical RGs, IPs, and neurons, respectively, in order to identify transcriptional differences in these cell populations. The genetically edited cortical CO cells showed very similar gene expression signatures with very few genes differentially expressed between themselves (Supplementary Fig. 4f–i), consistent with the previous observation of a lack of differences in the cellular phenotype, the proteome, and the secretome. We, therefore, pooled the cells derived from E370K and Y366Lfs COs (mutant cells, MUT). We identified genes that were up- and down-regulated in mutant compared to control CO cell populations (Supplementary Fig. 4f, j and Supplementary Data 1). Several genes were commonly differentially expressed, such as general RG markers—*FABP7, ID4, MDK, PTPRZ1*, and *PTN*—which were dysregulated specifically in mutant RGs (Supplementary Fig. 4k), suggesting that the RG population was mainly affected in the mutant COs. In addition, we observed that loci involved in NOTCH (*HES1, HES5, HEY1, DLL1*) and WNT (*GSK3B, WNT7B*) signaling pathways were differentially regulated in mutant cells (Supplementary Fig. 4k). These data potentially explain the role of *LGALS3BP* in neuronal differentiation of human NPCs and the non-cell-autonomous phenotype observed on the mislocalization of aRGs vs BPs. In addition, it was recently shown that several Wnt proteins have a different expression pattern throughout development and specifically, Wnt7b is important to respecify apical progenitors in mice changing their fate restrictions[51], thus, we speculate that changes in the transcriptional profile of *LGALS3BP* mutant cells may affect their specification and fate, and consequently their position.

Finally, focusing only on the RG population, we performed GO analysis for the most significant and enriched biological processes (Fig. 4i). Similar to the analysis in proteome and secretome, the highest processes affected specifically on RGs were related to cellular component organization, cellular process, and cellular localization. In particular, cellular component dysregulated processes were related to extracellular exosomes, extracellular organelle, and extracellular vesicles (Fig. 4i). On the contrary, similar analysis on the DEGs in IP and N did not reveal any

relevance in extracellular exosomes. Taken together, our data suggest that manipulation of LGALS3BP expression in human brain cells changes the extracellular niche of the cells, and more importantly that the major cell type affected are the RGs.

**Secreted factors rescue mutant COs' phenotype.** Having established that proper LGALS3BP expression influences the extracellular environment of human RG which regulates their delamination and in turn human cortical development, we wanted to demonstrate the specific action of LGALS3BP on these processes and tested whether *LGALS3BP* alone could rescue the phenotype observed in the mutant COs. To that end, we overexpressed *LGALS3BP* in control, E370K, and Y366Lfs COs and assessed the integrity of the apical belt and the neuronal positioning (Supplementary Fig. 5a–g, h–n). Upon *LGALS3BP* overexpression, the apical belt of the mutant COs clearly exhibited a patchy morphology in the electroporated area (Supplementary Fig. 5g). In addition, the forced expression of *LGALS3BP* in mutant COs could fully rescue the ectopic neuronal phenotype (Supplementary Fig. 5n), suggesting that proper expression of *LGALS3BP* is important for the positioning of neurons in the developing human cortex and elucidate the phenotype observed in individuals with *LGALS3BP* variations or mutant COs.

Then, taking into account the major differences in the extracellular environment observed at the protein level and the dysregulated genes in human RGs, we sought to test the extrinsic influence of LGALS3BP and specifically whether the secreted environment in control COs can influence the development of the genetically edited COs. To that end, E370K or Y366Lfs COs were treated with control COs' medium from 16 to 60d of culture (Fig. 5a). The medium switch was performed every day from COs generated from the same batch and the comparison was performed with COs cultured in their own medium. Analysis of the number of apically divided PH3+ cells (Fig. 5b–d), the thickness of the apical belt based on the phalloidin staining (Fig. 5e–g), and the number of ventricles with ectopic NEUN+ cells (Fig. 5h–j) showed that medium generated from control COs can completely rescue the phenotype observed in mutant COs. All the above strongly suggest that the secreted environment of human RGs is of great importance for their proper localization and proper human cortical development.

**LGALS3BP regulates location of NPCs and cortical development.** Given that the developing mouse cortex has a limited amount of BPs, is not as expanded as the human cortex and lacks folds, and that *Lgals3bp* exhibits negligible expression in the developing mouse cortex[18,32,52], we next investigated whether forced expression of the human gene isoform of *LGALS3BP* can promote the delamination of aRGs into BPs in the mouse model in vivo. Our hypothesis was based on the fact that previous

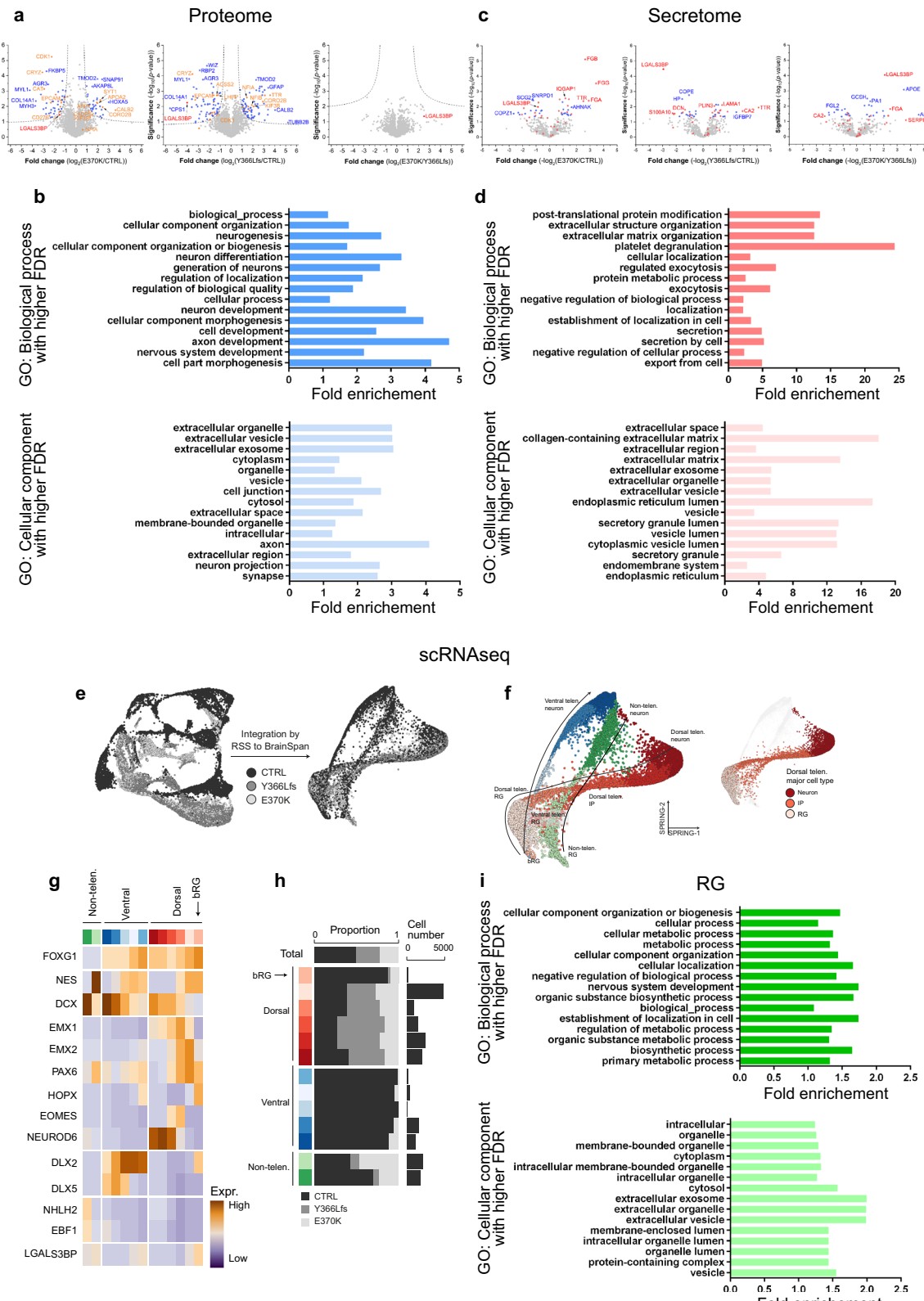

publications have shown that other human-specific or -enriched genes may be able to promote increased basally located progenitors and cortical folding in mice even though this normally does not happen[15,18,21]. To that end, we electroporated *LGALS3BP* in the developing mouse cortex at embryonic day 13 (E13) and analyzed the embryos at E16 (3dpe). The distribution of GFP+ electroporated cells did not change when *LGALS3BP*

overexpression was compared with control (Supplementary Fig. 6a–c), suggesting that the presence of the human gene caused no major morphological differences during this time window. We did, however, observe differences in the relative position of the different NPC types. Specifically, Pax6+ aRGs, Hopx+ (expressed in aRGs in mice and in bRGs in humans), and Tbr2+ BPs were ectopically located in basal areas, compared to their positioning

**Fig. 4 Proteomics, secretomics, and scRNA sequencing reveal altered signatures of the mutant cells in COs and cell non-autonomous effect of** *LGALS3BP* **function. a, c** Volcano plots illustrating the fold change of protein expression in the whole proteome (**a**) and in secretome (**c**) analysis in control or mutant COs at 60d. **b, d** GO Term analysis on the differentially regulated protein in the proteome (**b**) or secretome (**d**) of control or mutant COs plotted with reverse FDR and their fold enrichment. Data revealed higher enrichment of proteins with functions that are highly relevant to cortical development such as neurogenesis, cell-substrate adhesion, extracellular matrix organization, and secretions via vesicles like exosomes. **e** UMAP embedding of 10X genomics-based scRNA-seq data generated for 2 CTRL (9290 single cells), 2 Y366Lfs (5199 single cells), and 2 E370K (4095 single cells) COs prior to data integration (left) and SPRING embedding of the data upon RSS integration (right). **f** Cell type annotation of organoid scRNA-seq data visualized in the integrated SPRING embedding (see panel **e**). Three major progenitor-to-neuron trajectories can be seen for the dorsal telencephalon (shades of red), the ventral telencephalon (shades of blue), and non-telencephalic cells (shades of green). **g** Heatmap showing the expression of genes used to define the different cell types found in COs. The sidebar on the left denotes the cell states along the three main developmental trajectories (dorsal telencephalon, red; ventral telencephalon, blue; non-telencephalic cells, green). **h** Barplots summarizing the relative proportion of cells for each of the three conditions (left), as well as the total number of cells (right) detected for each cell state. **i** GO Term analysis on the differentially regulated genes in the RG population as where identified form scRNA-seq analysis form control and mutant COs plotted with reverse FDR and their fold enrichment. Data revealed higher enrichment of genes with cellular functions involved in extracellular exosomes, extracellular organelle and extracellular vesicles. See also Supplementary Fig. 4.
Abbreviations: CTRL: control, GO: gene ontology, FDR: false discovery rate, RG: radial glial cells, bRG: basal radial glial cells, IP: intermediate progenitor, Telen.: telencephalon, Non-telen.: non-telencephalic, Expr.: expression.

under control conditions (Fig. 6a–d, f–i, k–n). Quantification of the total number of Pax6+, Hopx+, and Tbr2+ cells showed no differences (Supplementary Fig. 6d–f). However, their percentages found in the VZ (BinA) were significantly smaller, while the percentage found basally in the SVZ in the intermediate zone (IZ, BinC) was doubled or more (Fig. 6e, j, o). Interestingly, many were not GFP+ indicating a cell non-autonomous phenotype. This suggests that in these cortices an oSVZ-like territory was formed, where bRG-like cells were found, reminiscent of observations in gyrencephalic species such as humans.

In addition, we also observed a small but obvious change (bend) in the band of Ctip2+ deep layer cortical neurons in 50% of the mice analyzed (Supplementary Fig. 6g–m). This curvature was located basally to the ectopic Pax6+ progenitors (Supplementary Fig. 6g–l). We, therefore, hypothesized that this could be the initiation of a rudimentary fold in an otherwise smooth mouse cortex. To test the hypothesis that the newly formed oSVZ-like territory contains functional bRGs, an additional analysis was performed at 6 and 13dpe. Forced expression of *LGALS3BP* for 6d in vivo promoted the expansion of the cortex including the generation of rudimentary gyri and sulci on the side of electroporation, while the contralateral hemisphere was completely smooth. Interestingly, these rudimentary folds were composed of both deep (Tbr1+) and upper (Satb2+) layer neurons and were maintained in the developing cortex until postnatal stage P7, 13dpe (Fig. 6p–w and Supplementary Fig. 6n–q). The expansion promoted upon *LGALS3BP* overexpression as found in 62% of the electroporated brains, 24% of which were classified as major folds and 48% as minor (Fig. 6x). The formation of folds was not observed in all the animals that were analyzed which may suggest that intrinsic mouse mechanisms may mask the phenotype, especially at later stages as the overexpression of LGALS3BP is transient, a possible consequence of the limitation of the mouse model to study folding. However, we observed the cortical expansion from as early as 3–13dpe significantly changing the way that the mouse cortex develops in vivo. To support these data and to investigate whether these folds have typical morphology as in the human cortex, we investigated the integrity of the basement membrane by immunostaining for laminin, an essential component of the basement membrane, and found that it was intact and encompassing the newly generated fold (Supplementary Fig. 6r–t') in most of the animals analyzed.

To investigate whether the *LGALS3BP* genetic variants influence the position of NPCs in vivo in mice or if this occurred only in a context where LGALS3BP is endogenously expressed, namely in the developing human cortex, we next investigated

whether these variants are sufficient to recapitulate the phenotype observed upon overexpression of the wild-type form of *LGALS3BP*. To that end, we electroporated the *LGALS3BP* variants found in individuals 1 and 2, E370K, and p.E294K (referred to as E294K), respectively, in the developing mouse cortex at embryonic day 13 (E13) and analyzed the embryos at E16 (3dpe). Immunostaining against LGALS3BP showed that all the variants, wt, E370K, and E294K, are expressed in the developing mouse cortex following electroporation and exhibit the same secreted pattern with dotty staining (Supplementary Fig. 7a–h'). Also, the distribution of GFP+ electroporated cells was indistinguishable between control, *LGALS3BP*, E370K, and E294K (Supplementary Fig. 7i). However, both E370K and E294K did not fully recapitulate the phenotype observed upon wt *LGALS3BP* overexpression, namely the generation of ectopic Hopx+ cells (Fig. 7a–e and Supplementary Fig. 7j). This suggests that, in contrast to the forced expression of the wt form of *LGALS3BP*, the variants are functionally insufficient to significantly affect NPC misposition in mice where, under normal conditions, aRGs are not generating increased number of BPs. In addition, only a small proportion of the mice electroporated with the two variants show the generation of folds (33% in E370K and 14% in E294K compared to 80% expressing wt *LGALS3BP*) (Fig. 7f–n). Since LGALS3BP regulates apical anchoring during COs development, we mechanistically investigated whether it acts similarly in vivo in RGs. Thus, immunostaining for β-catenin was performed after *LGALS3BP* overexpression in mice at 3dpe (Fig. 7o–r). At the site of electroporation, but not in adjacent regions or control electroporation, apical β-catenin was reduced (arrowheads) at almost 75% of the section analyzed (Fig. 7w). On the contrary, overexpression of the two *LGALS3BP* de novo variants, E370K and E294K, could not recapitulate this phenotype at full (Fig. 7s–w). Why the mutant *LGALS3BP* variants cannot recapitulate the phenotype observed when the wt form is introduced in the developing cortex of mouse can be attributed to the fact that, under physiological conditions, *LGALS3BP* is not expressed in the mouse. Thus, the overexpression of a less functional variant of the gene does not change the physiological expression levels which in turn explains the loss of the phenotype.

Overall, the above data suggest that forced expression of the human version of *LGALS3BP* in a context where it is normally not expressed, influences the anchoring of NPCs to the apical side, thus their position in the developing cortex and the generation of an oSVZ-like territory. Eventually, it regulates neuronal output and expansion during corticogenesis, including local gyrification in mice, which can adopt properties ascribed to the human context. On the contrary, the genetic variations in the

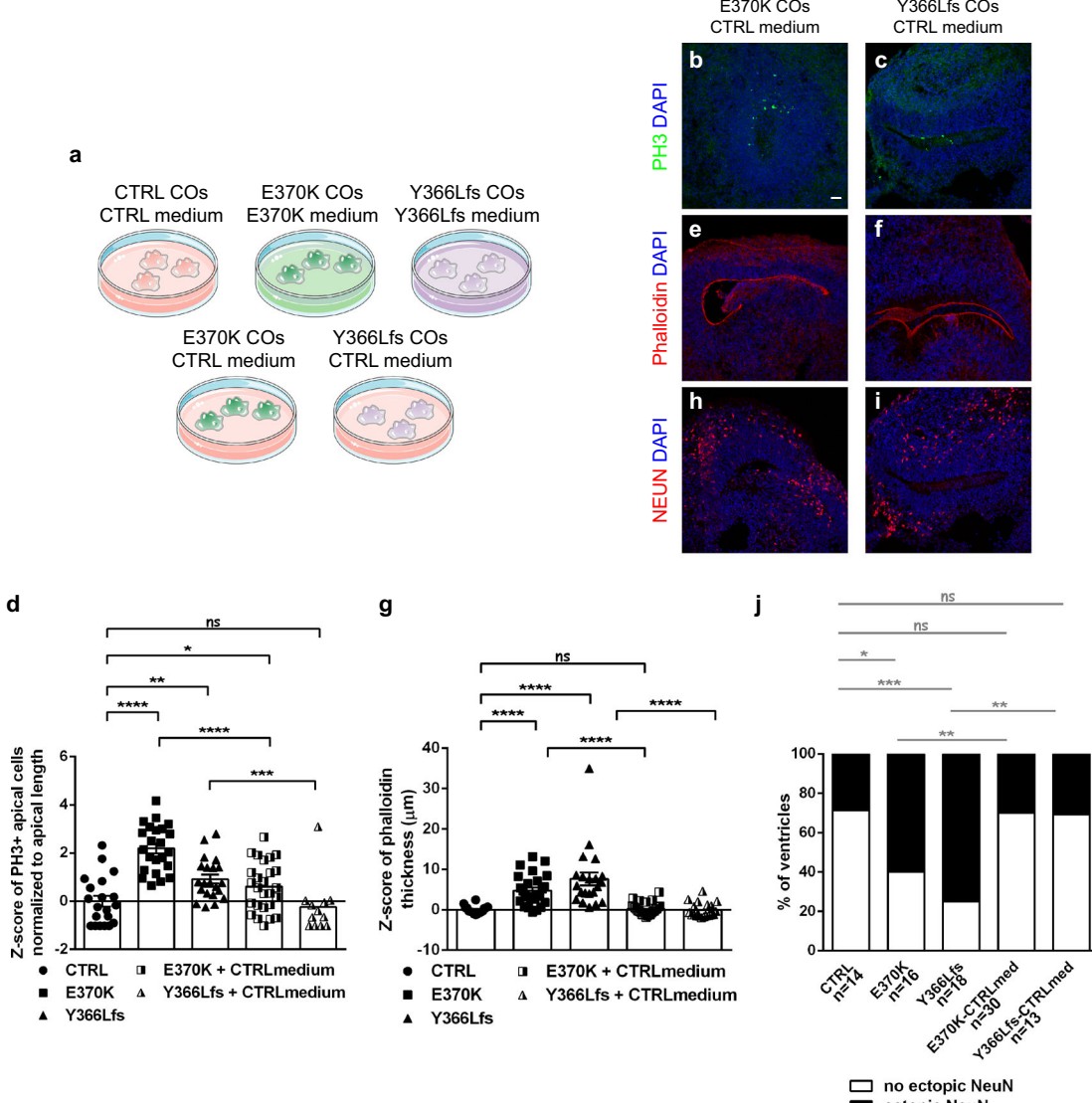

**Fig. 5 Proper protein secretion mediates LGALS3BP function. a** Scheme showing the experimental procedure of the switch medium experiment in COs. **b**, **c**, **e**, **f**, **h**, **i** Micrographs of sections of E370K or Y366Lfs COs treated with control condition medium and immunostained as depicted in the panels. **d**, **g** Quantification of the percentage of organoids with apically dividing PH3+ cells (**d**) or the phalloidin thickness of COs upon incubation of control, E370K, or Y366Lfs mutant COs with CTRL, E370K, or Y366Lfs medium as indicated in the panels. Data are shown as Z-scores ±SEM, statistical significance was based on two-tailed Mann-Whitney U test *$p < 0.05$, **$p < 0.01$, ***$p < 0.001$, ****$p < 0.0001$, batches = 1, organoids = 9, ventricles = 16. In (**d**), $p = 0.0026$ in ctrlvsY366Lfs, $p = 0.0357$ in ctrlvsE370K+ctrlmedium, $p = 0.0002$ in Y366LfsvsY366Lfs+ctrlmedium. In (**g**), $p = 0.0010$ in ctrlvsY366Lfs, $p = 0.0384$ in ctrlvsE370K+ctrlmedium. **j** Quantification of the percentage of organoids with ectopic neurons upon incubation of control, E370K, or Y366Lfs mutant COs with CTRL, E370K, or Y366Lfs medium as indicated in the panels. Data are shown in percentages, statistical significance was based on one-tailed exact binomial test *$p < 0.05$, **$p < 0.01$, ***$p < 0.001$, batches = 1, organoids = 9, ventricles = 16. In (**j**), $p = 0.0376$ in ctrlvsE370K, $p = 0.0002$ in ctrlvsY366Lfs, $p = 0.0012$ in E370KvsE370K+ctrlmed, $p = 0.0010$ in Y366LfsvsY366Lfs+ctrlmed. Scale bar: 30 μm. See also Supplementary Fig. 5. Abbreviations: CTRL: control.

*LGALS3BP* gene, which possibly act as dominant-negative, fail to promote the positioning of the NPCs in basal locations and eventually proper cortical expansion in a context where the endogenous *Lgals3BP* is not detectable.

**LGALS3BP is secreted via vesicles mediating function in mice.** Our data suggest that LGALS3BP, a secreted protein that was found in vesicles in the size of exosomes, changes the extracellular environment during human cortical development modulating its proper formation. In addition, scRNA-seq, proteomic and secretomic analyses pinpointed the role of extracellular exosomes in RGs as one of the most dysregulated processes. Finally, further

analysis in the DEGs from scRNA-seq of control and mutant CO cells, showed that some of the transmembrane proteins characterizing exosomes, such as CD81, CD82, and CD24 are differentially regulated in the RG population (Fig. 8a and Supplementary Data 1). Thus, we sought to investigate the role of EVs in cortical development ex vivo.

We were able to detect LGALS3BP in EVs isolated from control COs (Fig. 8b). To demonstrate that the function of LGALS3BP in influencing the position of NPCs and neurons is mediated by its content in secreted vesicles, we first produced EVs overexpressing either HA alone or in frame with wt *LGALS3BP* (HA-LGALS3BP), or the two genetic variants of *LGALS3BP* found in individuals 1 and 2 (HA-E370K and HA-E294K,

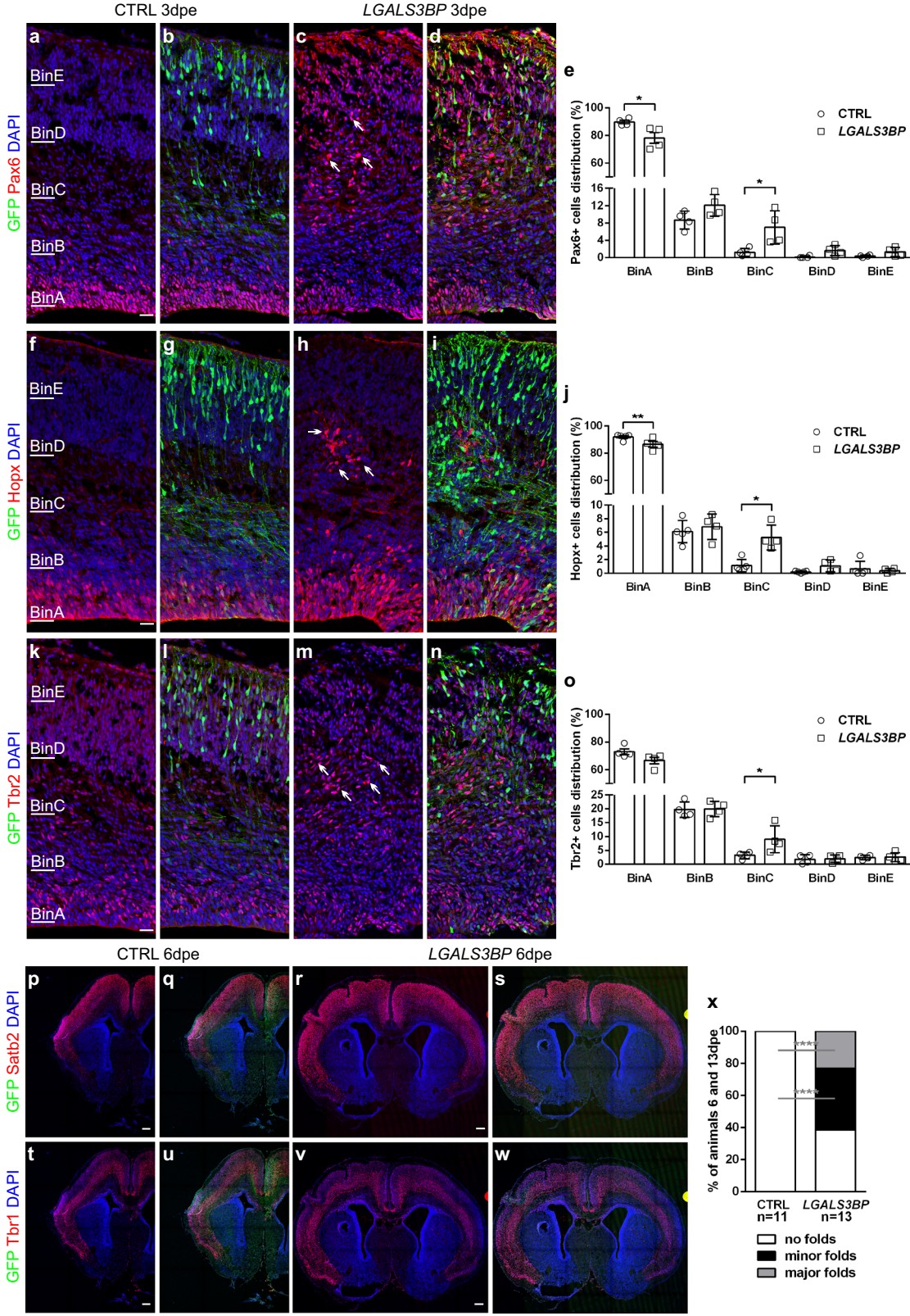

respectively) by overexpressing the constructs in SH-SY5Y cells and isolating the EVs from their culture medium. Since LGALS3BP has been found in circulating exosomes[40], we first tested that the EVs produced have a size between 100 and 150 nm, the typical size of exosomes (Supplementary Fig. 8a), and that all the different LGALS3BP variants are present in EVs (Fig. 8b). Then, we treated organotypic mouse slices isolated from

E13 mouse brains with the 4 collections of purified EVs. Control organotypic slices without EV treatment were also analyzed in parallel. Three days after the treatment, the mouse slices were fixed and the number of basally located Hopx+ progenitors and the position of Satb2+ neurons was assessed (Fig. 8c–g', i–m'). Mouse slices treated with HA-containing EVs did not differ from the non-treated slices in any of the parameters measured.

**Fig. 6 Overexpression of *LGALS3BP* in mouse brain results in changes in the progenitors' position and folded cortex. a–d, f–i, k–n** Micrographs of sections of mice immunostained as depicted in the panels after electroporation of control or *LGALS3BP* at embryonic day 13 and analyzed 3dpe. **e, j, o** Quantification of the distribution of the Pax6+, Hopx+ or Tbr2+ cells in the developing cortex. Data are shown mean ± SEM. Statistical significance was based on two-tailed Mann-Whitney *U* test *$p < 0.05$, **$p < 0.01$, $n = 4$ independently processed animals. In (**e**), $p = 0.0286$ in ctrlvsLGALS3BP in BinA, $p = 0.0286$ in ctrlvsLGALS3BP in BinC. In (**j**), $p = 0.0031$ in ctrlvsLGALS3BP in BinA, $p = 0.0159$ in ctrlvsLGALS3BP in BinC. In (**o**), $p = 0.0286$ in ctrlvsLGALS3BP in BinC. **p–w** Micrographs of sections of mice immunostained as depicted in the panels after electroporation of control or *LGALS3BP* at embryonic day 13 and analyzed 6dpe. *LGALS3BP* overexpression resulted in the formation of fold-like structures, which include deep (Tbr1) and upper (Satb2) layer neurons. **x** Quantification of the percentage of animals that had no folds (white), minor folds (black) or major folds (gray). Statistical significance was based on one-tailed exact binomial test **$p < 0.01$ ****$p < 0.0001$. Scale bar: 30 μm in (**a–d, f–i, k–n**), 200 μm in (**p–w**). See also Supplementary Fig. 6. Abbreviations: CTRL: control, dpe: days post electroporation.

Interestingly, slices treated with EVs containing HA-LGALS3BP, but not those treated with EVs containing the HA-E370K or HA-E294K variants, showed significantly increased number of basal Hopx+ cells (Fig. 8h), recapitulating the specific phenotype observed upon overexpression of the different LGALS3BP variants by in utero electroporation. In addition, mouse slices treated with variants of *LGALS3BP* showed the different location of neuronal clusters, observed ectopically in more apical locations, compared to control conditions (Fig. 8n) reminiscing the PH phenotype observed in patients.

Taken together, these results suggest LGALS3BP as a key protein in RGs' delamination and function, and in the regulation of human neurogenesis, cortical expansion, and the appearance of cortical gyrification. We propose that LGALS3BP is mostly produced by NPCs early during human neurogenesis, it is secreted via EVs and modulates the extracellular environment leading to the loosening of the apical anchoring and delamination of NPCs. Finally, this cascade of events may be responsible for the correct positioning of some neurons (Fig. 8o and Supplementary Fig. 8b).

## Discussion

The evolutionary expansion of the human cerebral cortex in comparison to that of other mammals occurred in concert with the emergence of marked differences in the cognitive abilities of our species. This expansion principally reflective of an increase in the total number of neurons in the cortex was previously proposed to be a consequence of an increased number of NPCs during early developmental stages[53,54]. Thus, genes regulating the pool of NPCs were considered important for determining human brain size. Another key step in the expansion of the brain was the emergence of a recently identified NPC type found in gyrencephalic species, the bRGs, and the expansion of the SVZ including IPs, except bRGs. Despite the essential role of BPs in human brain development, the mechanisms involved in their generation and function are still poorly understood. Here, we describe the role of *LGALS3BP*, a secreted protein that is part of the extracellular environment and interacts with several ECM proteins. *LGALS3BP* was depicted initially as one of the few genes defining the molecular identity of human bRGs[33], with comparably low levels of expression in the mouse cortex[18,24,32]. By performing a thorough expression analysis of LGALS3BP, we show that its expression is enriched in gyrencephalic species[17,34,35,37], it characterizes the major human NPC populations, namely aRGs, bRGs, IPs, its expression arises after aRG markers but before BP markers and interestingly, it is secreted. These data suggest that *LGALS3BP* may be one of the earliest genes regulating apical to basal NPC specification, placing it as a key player in the evolution of the human cerebral cortex. A few other genes have recently been described as key players: the primate-specific isoform of *PLEKHG6*, which regulates the number of apical versus basal NPCs[11], the *NOTCH2NL* paralogues which are a newly identified hominid-specific group of duplicated genes shown to promote the expansion of cortical progenitors in humans[19], *ARHGAP11B*, a human-specific duplicated gene, *HOPX* a bRG-specific gene promoting NPCs amplification and bRG generation in mice[18,21,33,55], TMEM14B a primate-specific gene promoting cortical expansion and folding by marking bRGs[56], and *Trnp1* a gene highly expressed in mice and which must be downregulated to regulate cerebral cortex expansion in gyrencephalic species[15]. Our work, together with previous studies, suggests that genes which are either human-specific or have a human-enriched expression pattern are essential for the evolution of gyrified species and the human cortex during development. Their importance is evident even in mice, a model system that lacks folds under normal conditions, suggesting that their expression should have been essential for the evolution of folds. It is therefore important to identify those human-specific or differentially expressed genes and to understand how cortical gyrification occurs.

Our findings show that *LGALS3BP* expression is important for the generation and positioning of apical and basal NPCs and neurons in humans, human-specific models, and mice. Since an important step during human neurogenesis is the migration of neuronal cells towards the developing cortex, aberrant neuronal migration can lead to ectopic neurons, resulting in cortical malformations, such as PH. A key cellular mechanism essential for correct neuronal migration is the morphology of aRGs. IPs and bRGs were recently proposed to be involved in malformations such as PH[9,11,57–59]. Individuals with genetic variations in *LGALS3BP* can exhibit not only PH but also changes in cortical thickness and the LGI. Thus, we propose that BPs' delamination mediated by LGALS3BP is critical for proper neuronal positioning and gyrification in humans. Furthermore, many cortical malformations are known to cause developmental delay and cognitive impairment[60]. Higher cortical functions, underlying cognitive processes, are one of the hallmarks of human evolution, suggesting that potential mechanisms regulating cortical evolution may interplay or be disrupted when cortical malformations occur. Therefore, candidate genes involved in both processes are of great importance. Interestingly, two other de novo mutations in *LGALS3BP* were identified in a cohort of individuals with autism spectrum disorder and a cohort with schizophrenia[61,62], and together with the fact that LGALS3BP is found in the recently developed SZDB database[63] and it is detected in the serum proteomes of patients with schizophrenia it suggests its potential contribution to the development of such disorders[64]. Moreover, the missense variant found in individual 1, exhibiting PH and GI abnormalities, is among the small number of non-conserved residues between human and macaque LGALS3BP[65]. Most of the non-conserved residues in macaque and the *LGALS3BP* human variants identified here are in exon 5, which is the region of the gene most intolerant to the acquisition of variation in healthy humans. Moreover, the expression of LGALS3BP in macaque-derived organoids shows a much lower number of cells expressing LGALS3BP compared to human-derived organoids[37]. These

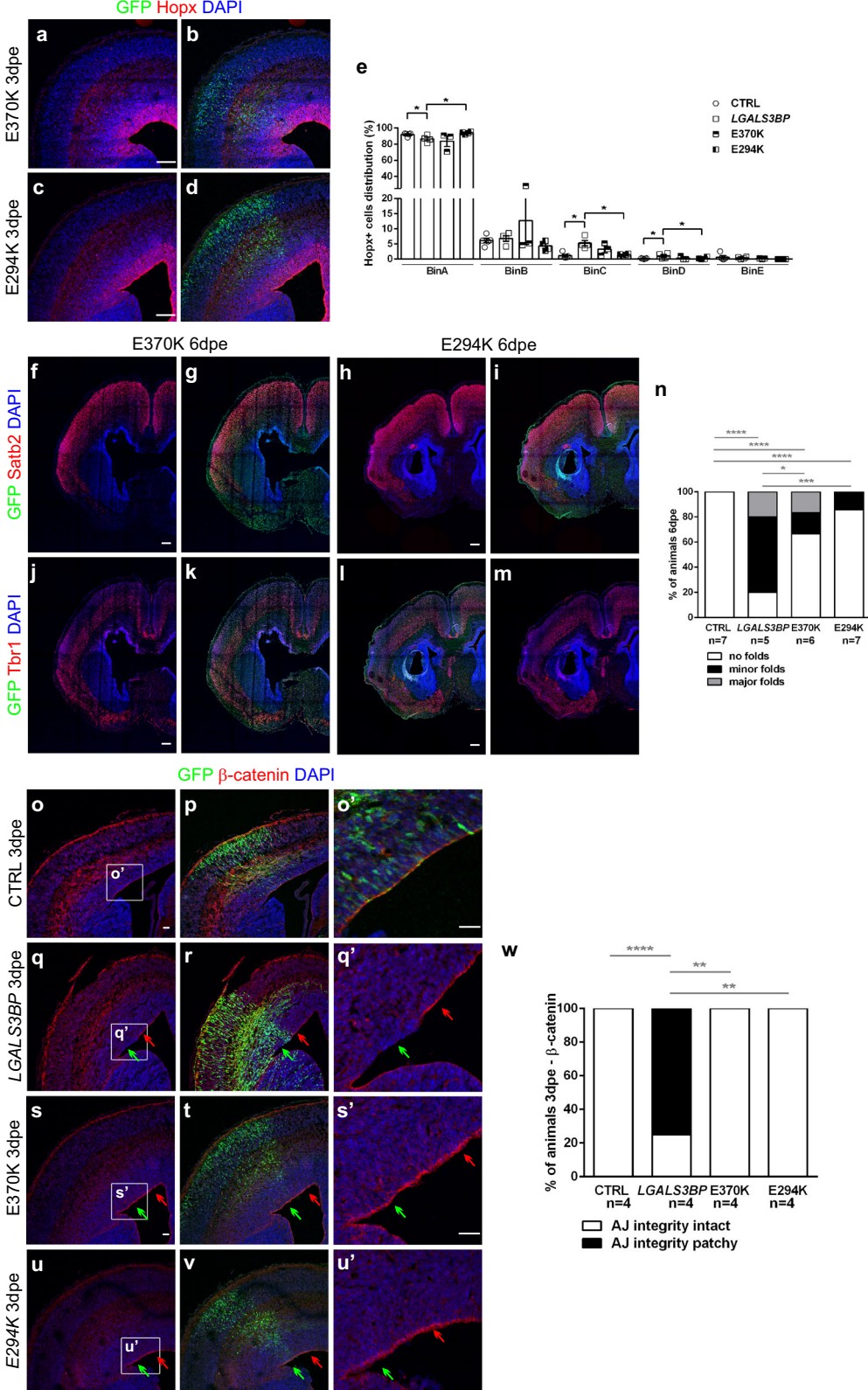

observations could suggest that this is a region of the gene which is of great importance for the acquisition of human-specific functions associated with gyrification.

Our data strongly indicate that the position of NPCs and neurons during human neurogenesis is mediated by the regulation of the extracellular niche where human NPCs are found, which modulates the proper anchoring of the apical progenitors to the VZ. We speculate that changes in the niche regulate the position of NPCs, and consequently the neuronal output during human cortical development. Here, we aim to bridge these mechanisms with the involvement of mechanical forces, which were recently suggested to be important for the proper folding of the human cortex[23]. Precedent exists for the importance of components of the ECM in corticogenesis. FLRTs, factors

**Fig. 7 Overexpression of the *LGALS3BP* variants in the mouse brain only partially recapitulates the phenotype. a–d** Micrographs of sections of mice immunostained for Hopx after electroporation of the E370K or the E294K variants of *LGALS3BP* at embryonic day 13 and analyzed 3dpe. **e** Quantification of the distribution of the Hopx+ cells in the developing cortex. Data are shown as mean ± SEM. Statistical significance was based on the two-tailed Mann-Whitney $U$ test *$p < 0.05$, $n = 5$ ctrl, 4 LGALS3BP, 3 E370K, 4 E294K independently processed animals. In (**e**), $p = 0.0317$ in ctrlvsLGALS3BP in BinA, $p = 0.0286$ in LGALS3BPvsE394K in BinA, $p = 0.0159$ in ctrlvsLGALS3BP in BinC, $p = 0.0286$ in LGALS3BPvsE394K in BinC. **f–m** Micrographs of sections of mice immunostained as depicted in the panels after electroporation of the E370K or the E294K variants of *LGALS3BP* at embryonic day 13 and analyzed 6dpe. Overexpression of the *LGALS3BP* variants cannot fully recapitulate the formation of fold-like structures, which the wt form of *LGALS3BP* generates. **n** Quantification of the percentage of animals that had no folds (white), minor folds (black) or major folds (gray). **o–v** Micrographs of sections of mice immunostained as indicated in the panels after electroporation of control (**o**, **p**), *LGALS3BP* (**q**, **r**), or the two *LGAS3BP* variants (**s–v**) at embryonic day 13 and analyzed 3dpe. Green arrows depict the electroporated area and red arrows the adjacent area. **w** Quantification of the percentage of mice with intact or patchy apical junction upon overexpression of the different forms of *LGALS3BP*. Statistical significance was based on the one-tailed exact binomial test, *$p < 0.05$, ***$p < 0.001$. In (**n**), $p = 0.0170$ in LGALS3BPvsE370K, $p = 0.0004$ in LGALS3BPvsE294K. In (**w**), $p = 0.0039$ in LGALS3BPvsE370K, $p = 0.0039$ in LGALS3BPvsE294K. Scale bar: 30 µm in (**a–d**), 200 µm in (**h–m**, **o–u'**). See also Supplementary Fig. 7. Abbreviations: CTRL: control, dpe: days post electroporation, LFQ: labeled free quantification.

involved in intercellular adhesion, as well as ECM components, such as HAPLN1, lumican, and collagen I, are examples of genes that have already been implicated in the development of the human cortex and gyrification[25,36]. In addition, members of the collagen family have been implicated in human cortical development and malformations[43,66]. In parallel to these, we show that LGALS3BP, another ECM-interacting protein, can also lead to defects in gyrification in individuals with *LGALS3BP* variants. More importantly, our work proposes that ECM, with LGALS3BP as a potential mediator of the ECM cues, can influence human cortical development at the cellular level. Interestingly, the crosstalk between ECM, LGALS3BP, and tetraspanins, which was proposed to regulate cell migration in cancer[31,67], could influence the rearrangement of the cytoskeleton and adhesion. Similarly, we show that LGALS3BP regulates the apical anchoring of NPCs in the VZ by influencing the function of apical polarity proteins and thus modulating BPs delamination. We also show that mutations in LGALS3BP change the composition of secreted proteins, and that it is present in EVs and that treatment with EVs expressing LGALS3BP is sufficient to reproduce the phenotype observed upon manipulation of its expression in vivo. We also point out that ECM components and exosomal receptors, such as tetraspanins with unique evolution-dependent expression patterns and/or specific abundance in the niche of human NPCs, are regulated by correct LGALS3BP expression and secretion. Mechanistically, we propose that the extracellular environment is affected by modulations of the expression of such proteins, capable of dynamically modifying the extracellular niche and resulting in loosening of the apical belt, delamination of BPs, and eventually changes in the neuronal output and cortical folding. Whether the changes in the niche of humans, when *LGALS3BP* genetic variations occur, are the consequence or the cause of their detachment/non-detachment from the apical belt is still uncertain. However, we favor the hypothesis that the niche influences the apical anchoring, this in turn influences NPC identity and fate as delaminated cells have their cell bodies in a different position and are exposed to different environmental cues.

Together, we suggest that *LGALS3BP* modulates the extracellular environment, regulates BPs delamination and thus, proper cortical formation and gyrification, as well as represent evidence that dysregulation of BPs position, morphology, and molecular identity can mechanistically underpin cortical malformations.

## Methods

**iPSC culture.** Induced pluripotent stem cells (iPSCs) reprogrammed from NuFF3-RQ human newborn foreskin feeder fibroblasts (GSC-3404, GlobalStem)[68]. Subjects gave consent for the generation of iPS cells (ISFi001-A). MTA approval for the use of this line of iPSCs was acquired. iPSCs were cultured on Matrigel (Corning) coated plates (Thermo Fisher, Waltham, MA, USA) in mTesR1 basic medium

supplemented with 1x mTesR1 supplement (STEMCELL Technologies, Vancouver, Canada) at 37 °C, 5% $CO_2$, and ambient oxygen level. Passaging was done by Accutase (STEMCELL Technologies) treatment.

**CRISPR genome editing for generation of mutant iPSCs lines.** For CRISPR genome editing for the generation of mutant COs, one control iPSC line was used to generate isogenic control and mutant lines. CrRNA was selected based on Geneious Prime, CCtop[69] and a guide design tool by Zhang lab (CRISPR.MI-T.EDU). The selected CrRNA (5′ AGTTCAACCTGTCCCTGTAC 3′) was assembled and in vitro transcribed into guideRNA (gRNA) by EnGen sgRNA Synthesis Kit, *Streptococcus pyogenes* (E3322S, NEB). The in vitro-transcribed guide was assembled into a gRNA-CAS9-NLS RNP complex by incubating the CAS9-NLS (M0646M, NEB) and in vitro-transcribed gRNA for 20 min at RT followed by electroporation into single cells of iPSC line using Amaxa 4D Nucleofector (Lonza). For generating the variation found in individual 1 (E370K) or the truncated/knockout exon5 line (Y366Lfs), targeting oligo (139 bp) was added into the gRNA-CAS9-NLS RNP prior to electroporation. Electroporated cells were plated onto a 10 cm Geltrex (A1413302, Thermo Fischer Scientific) coated dish supplemented with mTesR1 (85850, STEMCELL Technologies) and Y-27632 (72308, STEMCELL Technologies). The electroporated cells were allowed to recover for 10d with daily media changes of mTeSR1 before single-cell live sorting into Geltrex-coated 96-well plates using CloneR (05888, STEMCELL Technologies). The iPSC colonies were screened using the protocol by Yusa K[70] with the primers as listed in Supplementary Data 5.

The resulting PCR product (644 bp) was cloned and Sanger sequenced using the NEB PCR Cloning Kit (E1202S, NEB) to screen the individual genomic alleles.

The manufacturer's recommendations/kit protocols were followed for all the kits/reagents mentioned above.

*Sequencing the genomic regions after targeting and for OFF targets.* Genomic DNA was isolated from the E370K and Y366Lfs iPSC lines using QIAamp DNA Mini Kit (51306, QIAGEN) using the manufacturer's recommendations. The purified genomic DNA was used as a template to amplify the individual targeted and the putative off-target genomic loci predicted by the aforementioned guide design online tools and software. All the Sanger sequencing was performed at the Sequencing Facility of the Max Planck Institute of Molecular Cell Biology and Genetics, Dresden, Germany.

The information regarding the CrRNA, the targeting oligo, and the top 4 exonic positions of the putative off-targets, primer sequences, and amplified product sizes that were sequenced in the knockout (KO), and the patient-specific iPSC lines are included in Supplementary Data 4.

### Constructs for LGALS3BP manipulation

*Generation of overexpression constructs for* LGALS3BP. To overexpress in COs and in mice, the *LGALS3BP* gene, the open reading frame of the human *LGALS3BP* gene, was cloned into the PCAGGS plasmid vector[71] using standard cloning methods. To overexpress *LGALS3BP* in SH-SY5Y cells, the open reading frame of human *LGALS3BP* was cloned into the HA pcDNA3.1 plasmid vector (genescript) in frame with HA. For overexpression of the two variants found in the individuals with cortical malformations, the *LGALS3BP* in the PCAGGS or the pcDNA3.1 plasmid vectors were used as a template for point mutagenesis using the Q5® Site-Directed Mutagenesis Kit (BioLabs). The primers used for this procedure are listed in Supplementary Data 5.

As control and overexpression, the empty PCAGGS or HA pcDNA3.1 plasmid vectors were used, respectively.

### Generation and analysis of COs

COs were generated as previously described[38]. Briefly, iPSCs were dissociated into single cells using StemPro Accutase Cell Dissociation Reagent (A1110501, Life Technologies) and plated in the concentration of

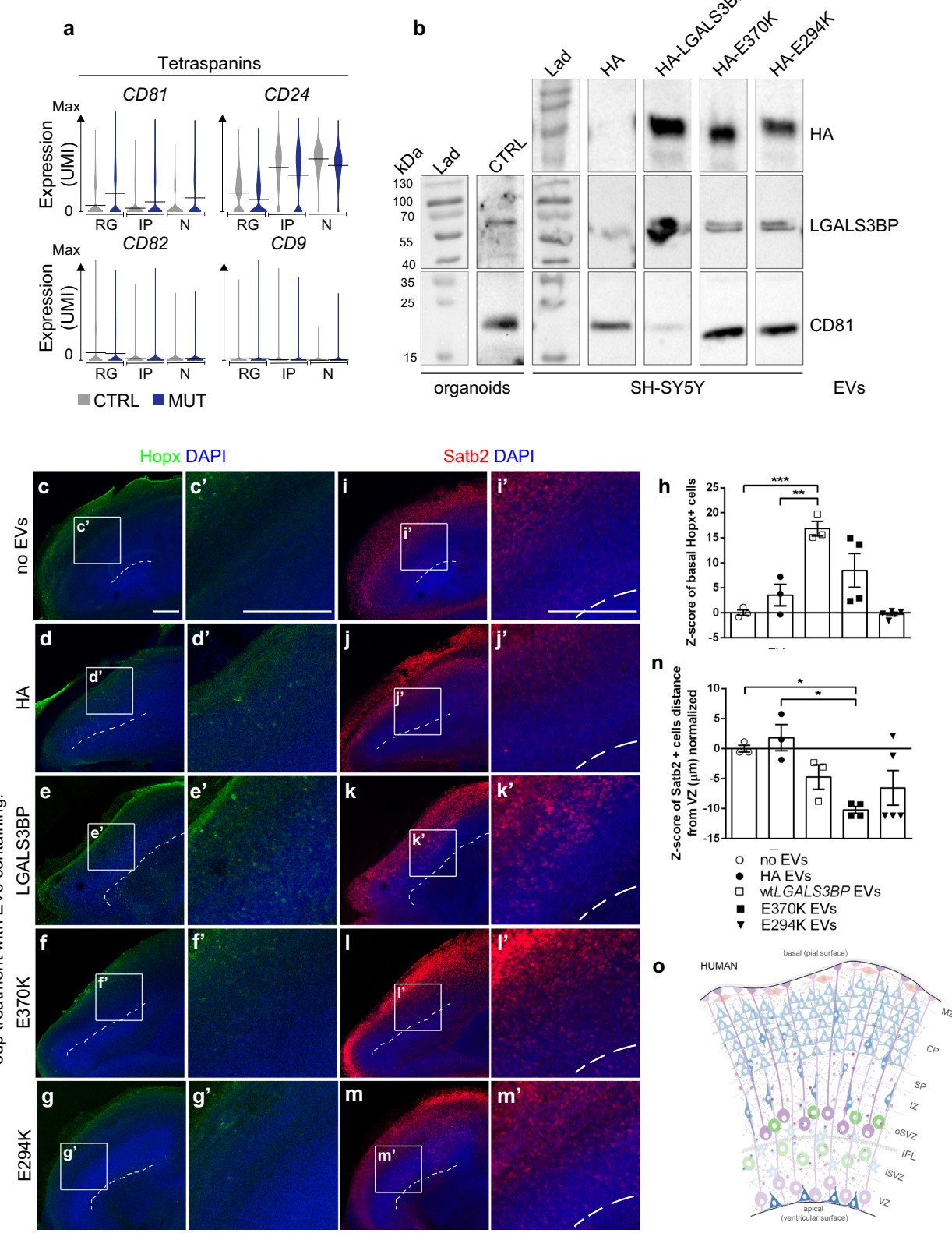

9000 single iPSCs/well into low-attachment 96-well tissue culture plates in hES medium (DMEM/F12GlutaMAX supplemented with 20% Knockout Serum Replacement, 3% ES-grade FBS, 1% nonessential amino acids, 0.1 mM 2-mercaptoethanol, 4 ng/mL bFGF, and 50 μM Rock inhibitor Y27632) for 6d in order to form embryoid bodies (EBs). Rock inhibitor Y27632 and bFGF were removed on the 4th day. On day 6, EBs were transferred into low-attachment 24-well plates in NIM medium (DMEM/F12GlutaMAX supplemented with 1:100 N2 supplement, 1% nonessential amino acids and 5 μg/mL Heparin) and cultured for additional 6d.

On day 12, EBs were embedded in Matrigel (Corning, 354234) drops and then they were transferred in 10 cm tissue culture plates in NDM minus A medium (DMEM/F12GlutaMAX and Neurobasal in ratio 1:1 supplemented with 1:100 N2 supplement 1:100 B27 without vitamin A, 0.5% nonessential amino acids, insulin 2.5 μg/mL, 1:100 Antibiotic-Antimycotic, and 50 μM 2-mercaptoethanol) in order to form COs. At 4d after Matrigel embedding, COs were transferred into an orbital shaker and cultured until electroporation in NDM plus A medium (DMEM/F12GlutaMAX and Neurobasal in ratio 1:1 supplemented with 1:100

**Fig. 8 LGALS3BP function is mediated by EVs. a** Violin plots showing the distribution of gene expression, which are up- or downregulated in mutant cells in scRNA-seq analysis. **b** Western blot analysis for the expression levels of HA, LGALS3BP, or CD81 antibodies performed in protein extracts isolated from EVs derived from control COs or SH-SY5Y cells upon electroporation of HA, HA-LGALS3BP, HA-E370K, or HA-E294K batches = 2, organoids = 6, cell batches = 3. Source data are provided as a Source Data file. **c–m'** Immunostaining as indicated in the panels of non-treated control mice sections or after incubation with SH-SY5Y-derived EVs overexpressing HA, HA-LGALS3BP, HA-E370K, or HA-E294K at embryonic day 13 and analyzed 3dpe. **h, n** Quantification of the percentage of basal Hopx+ cells (**h**) or the distance from the ventricular surface where the first neuronal positive cells are found (**n**) after treatment of control mouse sections with EVs overexpressing HA, HA-LGALS3BP, HA-E370K, or HA-E294K. Data are shown as Z-scores ±SEM. Statistical significance was based on one-way ANOVA and Turkey's multiple comparison test *$p < 0.05$, **$p < 0.01$, ***$p < 0.001$, $n = 3$ mice in no EVs, HA EVs, wt-LGALS3BP EVs, $n = 4$ mice in E370K EVs, $n = 5$ mice in E294K EVs. In (**h**), $p = 0.0002$. In (**n**), $p = 0.0142$. Scale bar: 200 μm. **o** Summary scheme showing that LGALS3BP is secreted via EVs protein, which upon secretion modulates the extracellular space and promotes changes in the distribution of the apical and basal progenitor cells which in turn regulates proper cortical development. Abbreviations: CTRL: control, MUT: mutant, UMI: Unique molecular identifiers, MAX: maximum, kDa: kilo Dalton, Lad: ladder, dpe: days post electroporation, EVs: extracellular vesicles.

N2 supplement 1:100 B27 with vitamin A, 0.5% nonessential amino acids, insulin 2.5 μg/mL, 1:100 antibiotic-antimycotic, and 50 μM 2-mercaptoethanol). During the whole period of CO generation, cells were kept at 37 °C, 5% $CO_2$, and ambient oxygen level with medium changes every other day. After transferring the COs onto the shaker, the medium was changed twice per week. COs were cultured up to 30, 40, 60, or 80d as indicated. For testing the effect of the secreted medium in COs cultures, COs from the same batch were cultured without medium switch until 16d. From 17 to 60d, the medium was switched every day by removing all the medium from the COs and adding half fresh medium and half condition medium from different genotype COs as described in the panels (Fig. 8 and Supplementary Fig. 8). COs were fixed using 4% PFA for 1 h at 4 °C, cryopreserved with 30% sucrose and stored at −20 °C. For immunofluorescence, 16 μm cryosections were prepared. For each experiment, many independent ventricles per CO from at least 3 different COs generated in 2–3 independent batches were analyzed.

**Generation of neural progenitor cells and neurons.** NPCs were generated as described previously[72] with modifications. In short, EBs were generated from iPSCs by plating colonies in suspension in the neural induction medium consisting of DMEM F12 with N2 and B27 supplements (Thermo Fisher). EBs were plated on polyornithine and laminin (Sigma Aldrich, St. Louis, MO, USA) coated dishes and cultured for 7d in the neural induction medium. Neural rosettes were manually picked using a stereological microscope (Nikon, Tokyo, Japan) and a P1000 tip, manually dissociated and further cultivated in neural progenitor medium (neural induction medium supplemented with bFGF at 20 ng/mL; Peprotech, Rocky Hill, NJ, USA). For passaging, the cells were dissociated using Accutase (STEMCELL Technologies) and split at a maximum ratio of 1:3.

**Electroporation of COs.** COs were kept in antibiotic-free conditions prior to electroporation. Electroporation was performed in COs at around 40d stages after the initial plating of the cells, and fixed 4dpe. During electroporation, COs were placed in an electroporation chamber (Harvard Apparatus, Holliston, MA, USA) under a stereoscope, and using a glass microcapillary of 1–2 μL, plasmid DNA was injected together with Fast Green (0.1%, Sigma) into different ventricles of the COs. COs were subsequently electroporated with 5 pulses applied at 80 V for 50 ms each, at 500 ms intervals (ECM830, Harvard Apparatus). Following electroporation, COs were kept for additional 24 h in antibiotics-free media and then changed into the normal media until fixation. COs were fixed using 4% PFA for 1 h at 4 °C, cryopreserved with 30% sucrose and stored at −20 °C. For immunofluorescence, 16 μm cryosections were prepared. For each experiment, many independent ventricles per CO from at least 3 different COs generated in 3 independent batches were analyzed.

**In utero electroporation.** Pregnant C57BL/6 mice were used as approved by the Government of Upper Bavaria under license number 55.2-1-54-2532-79-2016 (we have complied with all relevant ethical regulations for the use of mice). Mice were housed in case of a maximum of 3 animals per cage in a room with a 12-hour light–dark cycle, room temperature (RT) 20–24 °C, and relative humidity 45–65%. Mice were anesthetized by intraperitoneal injection of saline solution containing fentanyl (0.05 mg per kg body weight), midazolam (5 mg per kg body weight), and medetomidine (0.5 mg per kg body weight) (Btm license number 4518395), and E13 embryos were electroporated as described[73]. In brief, plasmids were mixed with Fast Green (2.5 mg/μL; Sigma) and injected at a concentration of 1 μg/μL. Anesthesia was terminated by injection of buprenorphine (0.1 mg per kg body weight), atipamezole (2.5 mg per kg body weight), and flumazenil (0.5 mg per kg body weight). Brains were fixed at 3dpe, 6dpe, or 13dpe in 4% PFA overnight. For immunofluorescence, 12 μm cryosections were prepared. For each experiment, at least 3 different mouse brains were analyzed.

**Human samples.** During the autopsy, human embryonic tissues at GW14–18 containing prefrontal cortex from one hemisphere were collected, stored on ice in DMEMF12 media and transported to the lab (ABM approval: PFS17-003) (we have complied with all relevant ethical regulations for the use of human embryonic tissues). Samples were then divided into smaller pieces and included in 4% low-gelling agarose with artificial cerebrospinal fluid[74]. DNA was injected through the gel, at the ventricular surface. A series of five electric pulses at 50 V for 50 ms at 1 s intervals was applied to the gel block. Tissue was then sliced at 300 μm and placed on Millicell-CM inserts (Millipore) in a cortical culture medium. After 6d of culture, tissues were fixed for 1 h in 4% PFA, detached from the insert, and stored at 4 °C in PBS. Brain slices were incubated in blocking solution (1X PBS, 0.3% Triton X-100, 2% normal donkey serum) for 2 h without agitation at RT. Slices were then incubated with primary antibody diluted in blocking solution overnight at 4 °C, washed, and incubated with secondary antibody diluted in blocking solution (Jackson ImmunoResearch) overnight at 4 °C. Incubation with 1X DAPI was included with the secondary antibody to visualize nuclei. Slices were washed for 1d before mounting.

We utilized whole-exome sequence data 65 trios (affected child and both parents) characterized and contributed by us in a previous study[11]. These individuals are also described in an additional study[43] where participants can be identified through the prefix 'pvhnz' in the cohort identifier table. In this same table, the sex of these participants can also be identified. All study participants were ascertained by physician referral, and phenotypes were assessed as sporadic based on patient and family interviews. The first patient was consented to participate under the University of Otago consent protocol. Parents or legally authorized representatives gave signed consent to participate in this study under protocols that were approved from either the Southern Regional Ethics Committee O03/016 or the New Zealand Ethics Committee MEC08/08/094 and was conducted in accordance with the criteria set by the Declaration of Helsinki. For the New Zealand-based study, general sharing of individual exome sequences was not approved on confidentiality grounds. The second patient, of whom clinical data were obtained, was identified within the DDD study[44]. This study has UK Research Ethics Committee approval (10/H0305/83), granted by the Cambridge South Research Ethics Committee and was conducted in accordance with the criteria set by the Declaration of Helsinki. All patients or their guardians provided written informed consent.

**Morphometric analysis on individual MRIs.** $T_1$-weighted MRI of the individual was processed according to a surface-based approach[46], to assess cortical thickness (CT)[75], LGI[47], surface area (SA)[76], and sulcal depth in different gyral and sulcal regions of the cortex[77]. Statistical analysis of morphometric data was performed comparing the individual with a template obtained by averaging 8 age-matched controls (ages: 15.6 ± 2.2 years, for subject 1; 1.1 ± 0.2 years, for subject 2). In order to detect differences in morphometric features for each hemisphere, the comparison was performed by a two-tailed paired $t$-test and cortical maps were clustered and corrected for multiple comparisons using a Monte-Carlo simulation (1000 iterations, Z-value = 1.3, cluster-wise $p$-value < 0.05). Cluster analysis in individual 1 (Fig. 2a) revealed reduced cortical thickness in the pre- and para-central gyri in both hemispheres. Clusters of increased LGI were found in the left lateral occipital gyrus and the anterior caudal segment of the right cingulate gyrus. LGI maps also disclosed simplified sulcal patterns, with reduced LGI, in the left superior-frontal and in the right medial orbitofrontal gyri. A statistically significant increase in the surface area was found bilaterally in rostral middle frontal and supramarginal gyri, in the right inferior-temporal gyrus, in the left insula, and in the precuneus. An increased sulcal depth was observed in the anterior part of the superior-temporal gyrus. Cluster analysis in individual 2 (Fig. 2a') revealed statistically significant diffuse abnormalities in the cortex, with increased thickness in cortical areas located between the precentral and the superior-frontal gyri in both hemispheres. Calculation of the LGI maps disclosed simplified sulcal patterns. Clusters of reduced LGI were found in the precentral, postcentral, and supramarginal gyri in both hemispheres. The area of the cortical surface appeared reduced in the left precuneus and the vertex of the postcentral gyrus. No alterations of the sulcal depth were found. Collectively, both individuals show morphometric changes in multiple cortical areas (Fig. 2a, a').

**Electroporation in SH-SY5Y cells, EVs isolation, and treatment in organotypic mouse sections**. SH-SY5Y cells were cultured in DMEM/F12GlutaMAX medium supplemented with 10% FBS and 1% antibiotics at 37 ℃, 5% $CO_2$, and ambient oxygen level. Passaging was done using trypsin/EDTA (Sigma) treatment. For the production of extracellular vesicles, 1d before the start of the experiment (day −1) cells were cultured in media with FBS depleted from bovine exosomes (FBS was ultracentrifuged in 100,000$g$ for 2 h and the supernatant was collected and used on the cells). On day 0, 2 million cells (80% confluency) were cultured in 15 cm plates in DMEM/F12GlutaMAX medium supplemented with 10% FBS (exosomes depleted) without antibiotics, and overexpression of HA, HA-LGALS3BP, HA-E370K, or HA-E294K, was performed via electroporation with 4 μgr of pcDNA3.1 expression construct using the Amaxa Nucleofector at the program G004. The following day (day 1), antibiotics were added to the medium. On day 2, the medium of the electroporated cells was collected and processed in a series of centrifugations (medium centrifugation in 300$g$ for 15 min, supernatant centrifugation in 2000$g$ for 10 min, supernatant centrifugation in 10,000$g$ for 30 min, supernatant centrifugation in 100,000$g$ for 120 min, and pellet wash with 1x PBS and centrifugation in 100,000$g$ for 60 min) to enrich for extracellular vesicles. For the nanoparticle tracking analysis (NTA), extracellular vesicle suspensions were diluted in PBS and analyzed using a Particle Metrix ZetaView® (Particle Metrix GmbH, Inning am Ammersee, Germany) equipped with a 520 nm laser. The manufacturer's default software settings for EVs were selected accordingly. For each measurement, samples were introduced manually and two cycles were performed by scanning at 11 discrete positions in the cell channel and capturing 60 frames per position (video setting: high). After capture, the videos were analyzed for particle size and concentration using the ZetaView Software 8.05.12 SP1. This analysis showed that 2 million cells produced 6,700000e + 008 particles per mL in control condition (HA), 6,100000e + 008 particles per mL in HA-LGALS3BP, 7,400000e + 008 particles per mL in HA-E370K, and 3,500000e + 008 particles per mL in HA-E294K. In all the conditions, the particle size was approximately around 150 nm. For the in vivo treatment of EVs on organotypic mouse slides, at the final step of EVs isolation the pellets were resuspended in 500 μL of mouse slice medium [DMEM/F12 (powder; Sigma; D2906), 2% Horse Serum (Gibco), 2% FCS (Gibco), 0.8% B27 (Gibco; 17504-044), 4% N2 (Gibco; 17502-048), and 1% P/S(Gibco)]. In parallel, control mouse brains from E13 mouse embryos were isolated, embedded into 3% of low-melting agarose, and cut in vibratome at 300 μm thick sections. The sections were placed on top of a cell culture insert (Millicell; PICMORG50) and cultured for 3d with 500 μL of mouse slice medium and 500 μL of the EVs-containing medium. After 3d, the sections were fixed with 4% PFA for 1 h and processed for immunofluorescence.

**Immunohistochemistry**. Immunostainings on sections were performed as described previously[78]. Briefly, sections were postfixed using 4% PFA for 10 min and permeabilized with 0.3% Triton X-100 (1332481001, Sigma-Aldrich) for 5 min. Sections were subsequently blocked with 0.1% TWEEN-20 (P9416, Sigma-Aldrich), 10% Normal Goat Serum, and 3% BSA. Immunostaining on floating sections was performed as follows: sections were placed in 24-well plates and washed with 1X PBS + 0.1% TWEEN, and then permeabilized and blocked with blocking solution containing 10% normal goat serum (NGS) (VEC-S-100 Biozol), 3% BSA, 0.3% Triton X. Primary and secondary antibodies were diluted in blocking solution. Nuclei were visualized using 0.5 μg/mL 4,6-diamidino-2-phenylindole (DAPI, Sigma Aldrich). Immunostained sections were analyzed using Leica laser-scanning microscope. F-ACTIN was visualized by incubation with Alexa Fluor 488-conjugated PHALLOIDIN (Thermo Fisher) according to the manufacturer's protocol. See the list of antibodies in Table 1.

**In situ hybridization**. Probes for in situ hybridization were generated as described[71]. Linearized in situ plasmids were in vitro transcribed using DIG NTP labeling mix, RNA polymerase T7 and T3, for sense and anti-sense probes, respectively, as well as RNAse inhibitor (Roche). In situ mRNA transcript detection was performed according to standard procedures.

**FACS analysis**. COs (30, 60, or 80d in culture) were collected for FACS analysis. Three to six samples were analyzed; every sample contained 3 individual COs. COs were enzymatically dissociated with Accutase, at 37 ℃ for 30 min. During incubation, every 10 min the COs were triturated with a P1000 pipette. After dissociation, samples were washed in PBS by centrifugation at 300$g$ for 5 min. The cell suspension was filtered through a 100 μm cell strainer and centrifuged at 300$g$ for 5 min, and the cells were fixed in 70% ice-cold Ethanol. After 1 h at −20 ℃, samples were centrifuged for 30 min at 500$g$ and then resuspended in 5 mL staining solution (PBS containing 1% FCS). After further centrifugation for 30 min at 500$g$, the cell pellet was resuspended in a staining solution containing anti-Pax6 antibody (1:250, see Table 1), anti-HOPX antibody (1:250, see Table 1), or anti-DCX antibody (1:250, see Table 1) and incubated for 30 min at 4 ℃. After washing in the staining solution, cells were resuspended in a staining solution containing Alexa-Fluor546 anti-guinea pig, AlexaFluor488 anti-rabbit (1:800), or AlexaFluor488 anti-mouse, secondary antibodies and incubated for 30 min at 4 ℃. After washing in PBS, cells were resuspended in PBS. FACS analysis was performed at a FACS

Aria (BD) in BD FACS Flow TM medium, with a nozzle diameter of 100 μm. For each run, 10,000 cells were analyzed.

*Gating strategy*. SSC-A/FSC-A gates were done to exclude cell debris and FSC-W/FSC-A to collect single cells (Source Data file). Specifically, the boundary between positive- and negative-staining cells was set according to each isotype control. Debris and aggregated cells were gated out by forward scatter and sideward scatter; single cells were gated out by FSC-W/FSC-A. Gating for fluorophores was done using samples stained with secondary antibody only. The flow rate was below 500 events/s.

**Western blot**. Cells or COs were lysed in lysis buffer (62.5 mM Tris-HCl, pH 6.8, 2% SDS, 10% sucrose in $H_2O$) with protease and phosphatase inhibitors (Roche, Basel, Switzerland) and approximately 20 μg of protein extracts were separated by SDS-PAGE with a 10% gel. Proteins were transferred to a nitrocellulose membrane (GE Healthcare, Chalfont St Giles, Buckinghamshire, Great Britain). For detection, membranes were incubated with primary antibodies overnight and with horse-radish peroxidase-labeled secondary antibodies at RT for 1 h, and afterward treated with ECL Western Blotting Detection Solution (Millipore, Billerica, MA, USA) to visualize bands. Bands were processed using Image Lab software.

**Antibodies**. See Table 1 for the list of antibodies used.

**Cell and tissue quantifications**. For the analysis of the electroporated COs and mice, the positive cells as indicated on the panels were quantified in at least 12 different CO ventricles from at least 6 COs generated in two independent preparations, or at least 6 mouse cortical sections from at least 4 embryos collected from 2 or more littermates. Analysis of the phenotype of the manipulation of *LGALS3BP* expression upon electroporation in COs was performed by always comparing COs grown in parallel in the same batch. Moreover, the findings were reproduced in different batches of organoids, grown at different times. Specifically, the analysis was performed as follows: only SOX2+ or HOPX+ cells found in the neurogenic zones were quantified, the number NEUN+ cells was quantified regardless if they reach the cortical plate or not. The number of puncta after immunostaining with LGALS3BP-specific antibody was performed with the puncta quantification plugin in ImageJ. For human fetal tissue, four samples ranging from GW14 to 18 were analyzed and a total of 7081 electroporated cells were counted. In the electroporation of these samples, the number of SOX2+ and NEUN+ cells found exclusively in the oSVZ were counted. For statistics, experiments were compared in a pared manner, to account for sample-to-sample variability. Analysis in the developing mouse cortices after electroporation was always performed and represented in the figures in the middle of the electroporation where the phenotype was strongest and more representative. For the total number of Hopx+, Pax6+, or Tbr2+ cells in the mouse cortex, pictures of the same size were used. For the distribution of the GFP+, Hopx+, Pax6+, or Tbr2+ cells in the mouse cortex, the cortex was subdivided into 5 equal bins—BinA corresponded to the apical side and BinE to the pial side of the cortex—in at least 6 sections from at least 4 embryos collected from 2 or more litters. To distinguish the degree of cortical folding in mice, we classified the folds in 6 or 13dpe as major when more than two sulci and gyri were observed, and as minor when only one gyrus was observed; while at 3dpe, mice with minor folds were characterized based on bulges of the Ctip2 neuronal layer. Analysis in the organotypic mouse slices after EVs incubation was always performed as follows: the number of Hopx+ cells found in basal from the VZ locations was quantified per cortex, the distance from the apical size of the cortex of the first cluster of NeuN+ cells with more than 5 cells was quantified in five different positions per cortex and normalized by the apical-to-pial surface length. In the control and mutant COs, the analysis was performed as follows: The CO size was measured by analyzing at least 12 COs from three independent preparations based on the surface of the COs. For the quantification of proliferating apical and basal NPCs, all PH3+ cells located either at the apical or at the basal side of the VZ were quantified from at least 3 COs and in at least 12 different ventricles, and then the cell counts were normalized by the length of the apical membrane side of the VZ. The analysis of non-delaminating HOPX+ cells was done by quantifying the number of ventricles which included more than 5 HOPX+ cells having an apical process as shown in Fig. 4i"–k". The analysis of the heterotopic NEUN+ cells in the VZ of control and mutant COs with or without *LGALS3BP* electroporation was performed by quantifying the number of ventricles, including more than 5 NEUN+ cells next to the ventricular lumen and at least 3 COs and at least 10 different ventricles were used for this analysis. The FABP7+ cells were calculated by counting the total number of FABP7+ nuclei found within the neurogenic zones of the COs. The apical belt thickness was identified using phalloidin, Pals1, or pan-cadherin immunostaining measuring at five different positions per germinal zone along the belt and normalizing the data by the length of the apical side of the VZ. The integrity of the apical junction was assessed by the continuous or patchy line of β-catenin or phalloidin staining in areas where the apical belt was well defined (ventricle form control of mutant organoids which did show properly defined staining of the apical anchoring markers at least in the dorsal part of the ventricles and in apical positions from the VZ were excluded from the analysis). All analyses were performed using ImageJ, Image Lab, Photoshop, or Illustrator and the

**Table 1 List of antibodies used.**

| Antigen | Dilution | Vendor | Catalog # | Lot # |
|---|---|---|---|---|
| PAX6 | 1:500 | Biolegend | PRB-278p | B244513 |
| LGALS3BP | 1:100 | eBioscience | BMS146 | – |
| SOX2 | 1:500 | Cell Signaling | 27485 | 2 |
| HOPX | 1:1000 | Sigma Aldrich | HPA030180 | B105571 |
| CTIP2 | 1:500 | Abcam | ab18465 | GR322373-4 |
| SATB2 | 1:500 | Abcam | Ab51502 | GR2075794 |
| TBR1 | 1:500 | Abcam | ab31940 | GR3217067-1 |
| LAMININ | 1:500 | Millipore | AB2034 | 2558444 |
| MAP2 | 1:500 | Sigma Aldrich | M4403 | 035MN4780V |
| PALS1 | 1:500 | Sigma Aldrich | 07-708 | H0907 |
| β-CATENIN | 1:500 | BD Biosciences | 610154 | 76645 |
| PH3 | 1:500 | Millipore | 06-570 | 3113883 |
| ARL13B | 1:200 | Proteintech | 17711-1-AP | – |
| GAPDH | 1:6000 | Millipore | CB1001 | 2896484 |
| GFP | 1:1000 | Aves Lab | GFP-1020 | 697986 |
| NESTIN | 1:200 | Millipore | MAB5326 | 3112610 |
| PAN-CADHERIN | | Sigma Aldrich | C1821 | 064M4764 |
| CD82 | 1:250 | Santa Cruz | sc-1087 | J2814 |
| TBR2 | 1:500 | Abcam | ab23345 | GR33045451 |
| PHALLOIDIN (ACTIN) | 1:40 | Thermo Fisher | A12381 | 1743642 |
| DoubleCortin (DCX) | 1:2000 | Millipore | AB2253 | 2787730 |
| NEUN | 1:500 | Millipore | MAB377 | 2742283 |
| KI67 | 1:500 | DAKO | M7248 | 20017551 |
| FABP7 | 1:1000 | Millipore | ABN14 | 3160120 |
| HA | 1:1000 | Santa Cruz | Sc-7392 | K1918 |
| CD81 | 1:250 | Santa Cruz | Sc-166029 | D0419 |
| Alexa Fluor® 647 Goat Anti-Mouse IgG (H + L) | 1:1000 | Life-Technologies | A-21235 | – |
| Alexa Fluor® 647 Goat Anti-Rabbit IgG (H + L) Antibody | 1:1000 | Life-Technologies | A-21244 | 2086730 |
| Alexa Fluor® 647 Goat Anti-Guinea Pig IgG (H + L), highly cross-adsorbed | 1:1000 | Life-Technologies | A-21450 | 2026140 |
| Alexa Fluor® 546 Goat Anti-Mouse IgG1 (γ1) | 1:1000 | Life-Technologies | A- 21123 | 1722393 |
| Alexa Fluor® 546 Goat Anti-Mouse IgG (H + L) | 1:1000 | Life-Technologies | A-11003 | – |
| Alexa Fluor® 546 Goat Anti-Rabbit IgG (H + L) | 1:1000 | Life-Technologies | A-11010 | 1971417 |
| Alexa Fluor® 546 Goat Anti-Rat IgG (H + L) | 1:1000 | Life-Technologies | A-11081 | – |
| Alexa Fluor® 488 Goat Anti-Chicken IgG (H + L) Antibody | 1:1000 | Life-Technologies | A-11039 | 2079383 |

appropriate statistical software package GraphPad for each analysis test as indicated in the figure legends. Nonparametric tests were chosen since they are more stringent. Details for the number of repetitions for each experiment are also included in the figure legends. Unless mentioned otherwise, measurements were taken from distinct samples. In graphs that are represented as Z-scores, Z-scores were generated from the raw data using the STANDARDIZE formula from the excel function where the ctrl mean and standard deviation were used.

**Bulk RNA-seq experiments**. COs (17, 23, 40, 70, 120, 140, or 158d in culture) were collected for bulk RNA extractions using the RNeasy Mini extraction kit (74104, QIAGEN) according to the manufacturer's instructions. Three replicates were analyzed per time point, with every sample containing 1–3 COs pooled. Sequencing libraries were prepared using the NEBNext® Ultra™ DNA Library Prep Kit for Illumina (E7370L, New England Biolabs) using ribosomal depletion as a selection method, and sequenced paired-end on an Illumina HiSeq4000 system at the Helmholtz Zentrum Core Facility (Munich, DE). Raw reads were processed using FastQC and cutadapt[79] and aligned using the STAR aligner[80]. The counts data were batch-corrected, normalized, and analyzed using the ImpulseDE2 framework[81].

**Dissociation of COs for scRNA-seq**. COs were dissociated as described in Miltenyi's Neural Tissue Dissociation protocol under sterile conditions. Samples were selected at day 60 and processed enzymatically to get a single-cell suspension. Firstly, each sample was cut into pieces and washed up to three times in 1x HBSS without $Ca^{2+}$ and $Mg^{2+}$ (HBSS w/o). Then samples were incubated for 15 min at 37 °C in the Enzyme Mix 1 with Papain (2 mL) and Enzyme Mix 2 (30 μL) was added once tissue started to disintegrate. In order to get sufficient dissociation, samples were gently triturated using 1 mL pipette tip and incubated for another 15 min at 37 °C. To evaluate the dissociation progress, the cell suspension was checked under the microscope. To generate single-cell suspension and remove clumps of cells, samples were applied to 30 and 20 μm diameter strainers and washed up to three times in HBSS w/o. Then samples were centrifuged at 300 RCF for 5 min and resuspended in HBSS w/o. Finally, the test of cell viability was performed using Trypan Blue solution (0.4%) while counting cells by an automatic

cell counter (Countess, Thermo Fisher). Single-cell suspension was used immediately for single-cell RNA-seq experiments.

**Single-cell RNA-seq experiments**. Single-cell RNA-seq experiments were performed on 6 organoids (2 organoids per condition CTRL (9290 single cells), Y366Lfs (5199 single cells), and E370K (4095 single cells)) using the Chromium 10X Genomics platform following the manual instructions of the Chromium Single Cell 3′ v2 Reagent Kit. In brief, 6000 cells per sample were loaded on a Single Cell A chip, following droplets with single cells and barcoded beads' generation, complementary DNA synthesis, preamplification, and library preparation. Sequencing-ready libraries with unique 10X sample index names were pooled at equal ratios and sequenced paired-end 26 × 8 × 100 base pairs on a HiSeq 2500 Illumina platform 2 lanes. Base-calling, adaptor trimming, and de-multiplexing were performed using 10X Genomics Cell Ranger 2.0 software. Data are shown in Supplementary Data 1.

**Processing, analysis, and graphic display of 10X genomics-based single-cell RNA-seq data**. As quality controls, cells with more than 6000 or less than 200 detected genes, as well as those with mitochondrial transcripts proportion higher than 10% were excluded. Additional filtering of cells was done based on the primary cell type prediction by using public human fetal brain sc-RNA-seq data[34]. Cells with predicted 'glycolysis' identity, as well as other cells in clusters (using Seurat, obtained for each CO separately) with more than 80% cells with predicted 'glycolysis' identity were excluded from the following analysis. Putative non-neuronal cells were further filtered, by only including cells with predicted radial glia, IP cells, excitatory neurons or inhibitory neurons, as well as other cells in clusters with more than 80% of predicted neuronal cells. The remaining cells were then represented as the normalized similarity spectrum to the gene expression levels of 237 fetal human brain samples in the BrainSpan database with bulk RNA-seq data. Pairwise distances between cells were calculated as the Pearson's correlation distance between their similarity spectrum representations, with which a kNN-network ($k = 20$) was constructed. The kNN-network was visualized using SPRING. The Walktrap community identification algorithm was applied to the kNN-network to identify cell clusters. Cells in the clusters with dorsal neural

progenitor or neuron signatures were combined as the dorsal forebrain lineage. The diffusion map algorithm was applied to cells in the dorsal forebrain lineage. The bRG signature score was calculated by the sum of expression levels of the five bRG marker genes which are also shown in Supplementary Fig. 4g (TNC, FAM107A, HOPX, PTPRZ1, LIFR) obtained from the Kriegstein Lab describing outer radial glia[33]. The ranks in DC1 were used as pseudotimes. Dorsal forebrain neural cells were then grouped into three groups based on their pseudotimes: dorsal forebrain radial glia, IPs, and neurons. To identify genes with DE between dorsal forebrain neural cells in control COs and *LGALS3BP* mutant COs, Wilcoxon's rank-sum test was applied to each gene for each of the three dorsal forebrain neuronal cell groups. In each cell group, DE genes were determined as genes with Bonferroni-corrected $p < 0.01$ and at least 1.3-fold change of average expression between the two conditions.

**Whole-proteome analysis from COs**. COs were washed twice with PBS and homogenized in ice-cold lysis buffer (1% (v/v) NP40, 1% (w/v) sodium deoxycholate, and 1 tablet protease inhibitor (cOmplete™, Mini, EDTA-free protease inhibitor cocktail, Roche) in 10 mL PBS) using ultrasonication at 40% intensity for 10 s. Lysis was done for 30 min at 4 °C while rotating. The insoluble fraction was pelleted (10 min, 14,000g, 4 °C) and protein concentration was determined by bicinchoninic acid assay (BCA, Carl Roth GmbH + Co.). For the next analysis, 200 μg of the total protein was used. Proteins were precipitated by the addition of 900 μL acetone and overnight incubation at −20 °C. The protein pellet was harvested by centrifugation at 9000g for 15 min at 4 °C and washed once with 0.5 mL of ice-cold methanol. Proteins were reconstituted in 200 μL digestion buffer (20 mM HEPES, pH 7.5, 7 M urea, 2 M thiourea), reduced (0.2 μL 1 M DTT, 45 min, 25 °C), and alkylated (2 μL, 30 min, 25 °C, in the dark). The alkylation reaction was quenched by the addition of 0.8 μL 1 M DTT and incubation for 30 min at 25 °C. Proteins were pre-digested with 1 μL LysC (Wako) at 25 °C for 4 h. Then, 600 μL 50 mM TEAB buffer was added and the proteins were digested overnight with 1.5 μL sequencing-grade trypsin (0.5 mg/mL, Promega) at 37 °C. The following day, the samples were acidified with 10 μL formic acid to a pH of 2–3. Peptides were desalted on 50 mg SepPak C18 cartridges (Waters Corp.) on a vacuum manifold. The cartridges were equilibrated with 1 mL acetonitrile, 1 mL 80% acetonitrile, and 3 mL 0.5% formic acid. The samples were loaded on the cartridges and subsequently washed with 5 mL 0.5% formic acid. The peptides were eluted two times with 250 μL 80% acetonitrile and 0.5% formic acid. Samples were combined and dried by lyophilization. Peptides were reconstituted in 30 μL 1% (v/v) formic acid, prepared for mass spectrometry by filtering through a membrane filter (Ultrafree-MC and –LC, Durapore PVDF-0.22 μm, Merck Millipore) and transferred into mass vials. Data are shown in Supplementary Data 2.

**Secretome analysis from COs**. For secretome analysis, culture media (5 mL) was collected from 60d COs after 4d of conditioning. The medium was cleared from dead cells and debris by centrifugation at 34g for 5 min and precipitated with methanol (20 mL) at −20 °C overnight. The precipitated proteins were collected by centrifugation at maximum speed for 10 min at 4 °C. The supernatant was removed. The protein pellet was reconstituted in the lysis buffer (1% (v/v) NP40, 1% (w/v) sodium deoxycholate, and protease inhibitors in PBS, 500 μL). Protein concentration was measured by BCA. Then, 300 μg of the total protein was used for further Wessel-Flügge precipitation. The total volume of 150 μL of secretome was completed with 600 μL of methanol, vortexed, 225 μL of chloroform, vortexed, and finally 450 μL of water, vortexed and sonicated in an ultrasound bath for 8 min. The upper phase was removed and discarded leaving the precipitated proteins at the interface intact. After the addition of the 450 μL of methanol, the mixture was centrifuged at maximum speed for 20 min at RT. The supernatant was removed and the resulting protein pellet was let dry on air. The protein pellet was reconstituted in 400 μL of digestion buffer and subsequently, 200 μL was used for reduction, alkylation, digestion by trypsin, desalting, and reconstitution of peptides as described above for whole-proteome analysis. Data are shown in Supplementary Data 3.

**Mass spectrometry**. MS analysis was performed on an Orbitrap Fusion instrument coupled to an Ultimate3000 Nano-HPLC via an electrospray easy source (all from Thermo Fisher Scientific). Samples were loaded on a 2 cm PepMap RSLC C18 trap column (particles 3 μm, 100 A, inner diameter 75 μm, Thermo Fisher Scientific) with 0.1% trifluoroacetic acid (TFA) and separated on a 50 cm PepMap RSLC C18 column (particles 2 μm, 100 A, inner diameter 75 μm, Thermo Fisher Scientific) constantly heated at 50 °C. The gradient was run from 5% to 32% acetonitrile, 0.1% formic acid during a 152 min method (7 min 5%, 105 min to 22%, 10 min to 32%, 10 min to 90%, 10 min wash at 90%, 10 min equilibration at 5%) at a flow rate of 300 nL/min. Instrument survey scans (m/z 300-1500) were acquired in the Orbitrap with a resolution of 120,000 at m/z 200 and the maximum injection time set to 50 ms (target value 2e5). Most-intense ions of charge states 2–7 were selected for fragmentation with high-energy collisional dissociation at a collision energy of 30%. The instrument was operated in top speed mode and spectra were acquired in the ion trap with the maximum injection time set to 50 ms (target value 1e4). The option to injections for all available parallelizable times was enabled. Dynamic exclusion of sequenced peptides was set to 60 s. Real-time mass calibration was based on internally generated fluoranthene ions. Data were acquired using Xcalibur software version 3.0sp2 (Thermo Fisher Scientific). Raw files were analyzed using MaxQuant software (1.6.2.10). Searches were performed against the Uniprot database for *Homo sapiens* (taxon identifier: 9606, 21st December 2018, including isoforms). At least two unique peptides were required for protein identification. False discovery rate determination was carried out using a decoy database and thresholds were set to 1% FDR both at a peptide-spectrum match and at protein levels.

Statistical analysis of the MaxQuant result table proteinGroups.txt was done with Perseus 1.5.1.6. Putative contaminants and reverse hits were removed. Labeled free quantification intensities were log2-transformed, hits with less than 3 valid values in each group were removed and −log10(p-values) were obtained by a two-sided two samples' Student's t-test over replicates with the initial significance level of $p = 0.05$ adjustment by the multiple testing correction methods of Benjamini and Hochberg (FDR = 0.05), the −log10 of p-values were plotted by volcano plot function. Final volcano plots were edited in Origin.

**GO Term analysis**. For GO Term analysis of proteins or genes significantly downregulated or upregulated in the proteome, secretome or scRNA-seq of control and mutant COs were used to identify overrepresented functional groups and single proteins/genes falling into those with Panther, choosing *Homo sapiens* as species and searched for enrichment of Biological processes or cellular component complete using Fisher's exact test and FDR threshold of 0.05.

**Reporting summary**. Further information on research design is available in the Nature Research Reporting Summary linked to this article.

## Data availability

All data supporting the findings described in this paper are available in the article and in the Supplementary Information, and from the corresponding authors upon reasonable request. Data illustrated in Supplementary Fig. 1a are obtained from[34] and cited in this paper. For the New Zealand-based study on human individuals, unrestricted sharing of individual exome sequences was not approved by the New Zealand Ethics Review Committee on confidentiality grounds. Please contact Prof. Stephen P. Robertson (co-corresponding author) for access request. A response will be given within 1 month upon request. The bulk RNA-seq data used in this study have been deposited in the Gene Expression Omnibus under accession number GSE181405. The scRNA-seq data used in this study have been deposited in the ArrayExpress under accession number E-MTAB-10485. The mass spectrometry proteomics data and mass spectrometry secretomic data have been deposited to the ProteomeXchange Consortium via the PRIDE[84] partner repository with the dataset identifier PXD015878. There is no restriction for data access in all the above. Source data are provided with this paper (Supplementary Data 6). Web resources used in this paper are listed here: DECIPHER, https://decipher.sanger.ac.uk/ scRNA-sequencing data from human fetal cortex https://cells.ucsc.edu/?ds=cortex-dev gnomAD Browser, v.r2.0.2, http://gnomad.broadinstitute.org/ Source data are provided with this paper.

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

## Acknowledgements

We thank the families participating in this study for their involvement. We thank all the members of the lab for fruitful discussions, particularly Ane-Cristina Ayo-Martin, Fabrizia Pipicelli, Carola Eggert, and Timucin Öztürk for technical support and comments on the manuscript, Christian Grätz and Prof. Dr. Michael W. Pfaffl for their help with the nanoparticle tracking analysis, and Jessica Keverne for proofreading the manuscript. The DDD study presents independent research commissioned by the Health Innovation Challenge Fund [grant number HICF-1009-003]. This study makes use of DECIPHER (http://decipher.sanger.ac.uk), which is funded by the Wellcome. See Fitzgerald et. al.[82] or www.ddduk.org/access.html for full acknowledgment. A full list of centers that contributed to the generation of the data is available from http://decipher.sanger.ac.uk and via email from decipher@sanger.ac.uk. Funding for their project was provided by the Wellcome Trust[83]. This work was supported by funding from the Max Planck Society (SC-CK-RDG-IYB), the Health Research Council of New Zealand and Curekids (SR), the Alexander von Humboldt Foundation (CC), the Philip Wrightson Postdoctoral Fellowship from the Neurological Foundation of New Zealand (ACO), the EU Seventh Framework Programme (FP7) under the project DESIRE grant agreement 602531 (RG).

## Author contributions

Conceptualization, C.K. and S.C.; Methodology, C.K., A.C.O. and A.B.; Investigation, C.K., A.C.O. A. Brazovskaja, Z.H., P.K., A.F.E., L.C. R.D.G., P.D.A., A. Belka, D.M., B.H., M.L., C.C. and I.Y.B.; Resources, S.K., G.F., E.B., F.E., R.G., A.D.B., S.A.S., B.T., S.P.R., and S.C.; Human genetics data interpretation and analysis: S.P.R.; Writing original draft, C.K.; Writing review & editing, C.K., S.P.R. and S.C; Visualization, S.C.; Supervision, S.C.; Funding acquisition, S.P.R. and S.C.

## Funding

## Competing interests

The authors declare no competing interests.
