## [Peer Review File. · Nature Communications]

REVIEWER COMMENTS

Reviewer #1 (Remarks to the Author):

Kyrousi et al., performs a significant amount of work in both in vitro and in vivo cortical development models to characterize the role of human LGALS3BP mutations, encoded by Galactin-3 binding protein, which interacts with members of the ECM. The role of LGALS3BP is well established in cancer biology but less is known during brain development. The main question to be addressed is understanding the mechanisms that contribute to the expanded cortical size and generation of gyri and sulci in humans. While increasing rounds of basal radial glial progenitor division has been described as a possible mechanism, this paper focuses on an extrinsic mechanism, namely the role of the extracellular matrix, in neocortical expansion and cortical folding. Therefore this paper is conceptually interesting and novel in this regard. In a nutshell, the authors provide data which support the following conclusions:

1. LGALS3BP is a human-enriched ECM molecule in neurogenic zones near NPCs shown by IHC and in situ hybridization of cortical organoids at different stages and is a secreted protein from cortical organoid vesicles.
2. iPSCs gene edited cortical organoids with patient specific LGALS3BP mutations exhibit misplacement of NPCs and neurons, including changes in in NPC divisions, radial glia markers, and other morphological changes, consistent with individuals carrying these variants with neurodevelopmental conditions showing cortical malformation.
3. LGALS3BP levels is underlying this phenotype as overexpression of LGALS3BP (both WT and mutant variants) is sufficient to misplace NPCs and neurons in mouse cortex development and slices of human fetal brain.
4. LGALS3BP secretion from exosomes is suggested to be the purported mechanism to alter the protein composition of the extracellular niche. But there is no direct evidence for this, this is the weakest part of the story, therefore I suggest this part to be removed or simplified as part of another paper.

Despite these strengths, there are also major weaknesses in the presentation of the data to fit a cohesive story. There is an overabundance of data, sometimes the organization gets unwieldy, and the results section is very long. It seems like there's two separate stories here, and it's not clear how the significant amount of 'omics data, e.g., transcriptome, proteome, secretome, come together to produce a mechanistic explanation for the cellular and phenotypic results.

Major comments:

1. To strengthen the conclusion that LGALS3BP is a human-enriched ECM molecule, the authors should provide the data, or clearly describe the previous work, that it is not expressed during mouse (or other species) cortical development. Also it is implied that the individuals with the de novo LGALS3BP mutations have higher levels of LGALS3BP, so does this mean they are gain of function mutations? What is the difference between the E370K and Y366Lfs mutations, in terms of more or less LGALS3BP secretion? Why do the authors study both; more compelling rationale in the paper should be described. The authors should directly examine the level of LGALS3BP in the medium of Ctrl vs mutant organoids and correlate it to the expression of LGALS3BP within the organoid tissue with the mutant phenotypes.
2. The authors examine many aspects of cellular phenotypes in cortical organoids – progenitor cell distribution, cell division, delamination, cortical thickness. But central to the story is the effect of LGALS3BP on the formation of gyri and sulci. While this is examined in the overexpression studies of the human genes in mouse, the authors do not directly examine folding in organoid culture. Perhaps

this is because there is no effect on folding or organoid morphology. The authors should be explicit about this because there seems like a disconnect here. The authors should also consider adding a table to summarize the many different cellular phenotypes. This gets unwieldy very quickly.

3. How do the authors interpret that overexpression of the WT LGALS3BP (but not mutant variant) in the mouse model does not result in major morphological differences in folding, but an effect on progenitor cell distribution? How would the mouse niche compensate for an ECM molecule it does not normally express?

Minor comments:

1. The format used for the references is confusing.
2. What do the authors mean by range and frequency in extended data figure 1b?
3. Figure 1a shows peak of expression of LGALS3BP at 120D. However, the authors chose to perform single cell RNA seq, proteome and secretome analysis at 60D rather than 120D. The authors should make a comment to clarify this.
4. The figure legend for Figure 1c is missing.
5. What was the sequencing depth of the single cell libraries?
6. Page 6, line#128-129, the authors write that LGALS3BP staining "confirms its presence in vesicles as recently reported". This sentence is slightly misleading as the authors have not performed staining with markers of vesicles. Moreover, it's not clear from the stainings whether LGALS3BP is present in the cells or around the cells?
7. The rationale for choosing 40d time point for overexpression studies is not very clear.
8. In figure 2d-f, the authors claim that there is a difference in the distribution of NPCs in the variant background and have performed immunostaining with PH3, a marker for dividing cells. However, to be sure that it's only the neural progenitor cells that are dividing, it might be better to add a marker of neural progenitor cells as well in this staining.
9. In figure 2g-h, it is not clear what do the authors mean by apical cells normalized to apical length and basal cells normalized to apical length.
10. The authors should clarify how they calculated the Z-score in their analysis.
11. Figure format/organization:
 - Images in the figures are good but the organization of the panels is confusing. The authors should strictly adhere to proper citation of the figures. In some figures later panels are discussed earlier in the results. For example, figure 1b is discussed before figure 1a. Similarly, figure 3s,t are discussed before 3b-e,o and so on and so forth.
 - Some graphs are small which makes it hard to evaluate the data (Fig.2b and c; Fig.4a and c)
 - Statistical data is not easy to see in some graphs (Fig.2l, Fig.3i, Fig.5j, Fig.6x, Fig.7w). It would be better if the asterisks signs are inserted above the bar graphs and not in between the data bars.
 - The sample size should be included in figure legends or figures.

Reviewer #2 (Remarks to the Author):

In this work, Kyrousi et al., examine the functional role of LGALS3BP in neocortical development using both human (organoid and patient clinical data) and mouse models. LGALS3BP was predicted to be of importance in cortical development due to the increased expression of this protein in human NPCs which is shown in this work using human organoids and primary human tissue. These authors identified genetic variants in LGALS3BP in three human individuals (E370K, E294K, E527G) and go on to show that some of this genetic variation, along with LGALS3BP overexpression, influences NPC delamination and positioning by acting extrinsically via exosome secretion. This is demonstrated using

a wide variety of techniques including immunostaining, proteomics, and single-cell RNA-sequencing (scRNA-seq). Finally, these authors show that LGALS3BP overexpression in the mouse cortex impacts NPC (and progeny) position in the cortex and induces folding, while the genetic variants are less able to do so. This is an interesting study that adds another gene/protein to the growing list which play human-specific roles in cortical development. The authors should be commended for undertaking such a diverse set of studies to understand the role of LGALS3BP and its genetic variants. I do, however, have a number of major and minor points which should be considered/addressed.

Major:

1) LGALS3BP expression in mouse versus human cortex isn't shown in Extended Data Fig 1b where the callout occurs on Line 106. The authors have experience with both mouse and human systems. This should be shown by directly comparing the expression of LGALS3BP in mouse and human cortex as this is the primary rationale for this work.

2) In Figure 2A and A' there is a really obvious difference between these two patients (and thus the variants). This is not mentioned in the manuscript text. Why is the difference between these variants so large and what was the rationale for selecting E370K (the more modest phenotype) for most of the rest of the experiments? Moreover, E294K (the more significant phenotype) has more modest effects in the mouse (Figure 7n, for example) compared to E370K. These points should, at minimum, be addressed with further discussion.

3) In Extended Figure 2E LGALS3BP doesn't appear to be reduced in the E370K samples while Y366Ifs seems to be much lower. Quantitative data should be provided to support the claim that both E370K and Y366Ifs samples have reduced LGALS3BP. Moreover, in Figure 4A, in the case of E370K LGALS3BP is not significantly different in the proteome data. This is consistent with the E370K data in Extended Figure 2E. Are all of these variants loss-of-function? Overall, the impacts of the different mutations need to be clarified/discussed in more detail.

4) In a number of cases in the organoid data, the images shown either seem 1) not to be similar to similar comparisons; or 2) the representative data shown doesn't seem to well-represent the quantitative data shown; or 3) it is unclear what is being shown in the figures. For example, A) in Fig 2i',j',k" impaired delamination is being described but it's not clear what is being shown in these panels; B) in Fig 3 c/d/c' these sections don't look like similar comparisons and it's unclear what is being shown here; why are their nuclei below the apical border?; and C) Extended Data Fig 3b-e'/f-i'/j-m' the images shown don't seem to represent the quantitative data and are not all like comparisons.

5) Loss of LGALS3BP causes centriole accumulation in cancer cells (Fogeron, Nat Comm, 2013) and centrioles are important for organizing cilia. In Fig. 3v-x ARL13B (a known centriole/cilia marker) seems to be increased in the mutant organoids. Cilia dysregulation is known to impact cortical development and cilia are important for responses to extrinsic cues. The authors should investigate whether the impacts of LGALS3BP mutations are due to dysregulated cilia.

6) Folding is shown to only occur in 62% of the mouse brains at 13 dpe using LGALS3BP overexpression. Why is there so much variability in the folding phenotype? Is it due to the number of electroporated cells per brain or the degree of overexpression or something else? This should be investigated and clarified.

Minor:

1) In Fig 1c-e, the in situ data is so low magnification it's not very clear what is being shown. Higher magnification images would be beneficial.

2) In Figure 5 there are no images of Ctrl COs with Ctrl medium and thus difficult to interpret these data.

3) In the scRNA-seq data in Figure 4, What is the contribution of individual organoids or were organoids combined? How robust are these data across different organoids? This should be clarified. Also, in Extended Data Fig 4 it's not clear how bRG are being defined and how signature scores are being computed (I don't see this in the methods).

4) The quantitative data in Figure 7n has no statistical testing and 7w has low N when using proportions; one more or less animals in each group could dramatically change these data.

5) Finally, in the HA-LGALS3BP blot in Figure 8b there is a blob where the LGALS3BP band would be. That is unfortunate, because it's not possible to tell if there is a band there or not. This is important because HA-LGALS3BP has the largest effect in the functional studies. Ideally, this should be replaced/repeated.

Reviewer #3 (Remarks to the Author):

In this manuscript the authors have investigated the functional role of LGALS3BP in brain development. Evidence showed that LGALS3BP is enriched in human neural progenitor cells, LGALS3BP mutations found in neurodevelopmental diseases were associated with cortical malformations and LGALS3BP controlled apical anchoring of human progenitor cells. Furthermore, a set of evidence (comparing wt vs LGALS3BP mutants) including the molecular signatures of the secretome from cerebral organoids (CO), the effect of CO medium on CO differentiation and the effect of exosome-enriched fractions (from SH-SY5Y cells) on organotypic mouse brain slices led the authors to propose that LGALS3BP would be secreted via exosomes, relevant for its biological function.

An impressive variety of complementary techniques and biological systems (humans, human-specific and mouse model) have been studied. In the literature, LGALS3BP has been more widely studied in cancer and only more recently it started to be investigated in the nervous system and neurological diseases. In view of this, results are very innovative and constitute a very significant advancement to elucidate the role of LGALS3BP in human corticogenesis.

The work is described with great detail and the results are in general of high quality and convincing. However, there are the following concerns:

1. Extracellular vesicles are released from cells and are heterogeneous (Cocozza et al PMID: 32649878); they include exosomes (endosomal origin; 50-150 nm diameter), microvesicles (budding of plasma membrane) and, more recently, exomeres (30-50 nm diameter) have been identified. Data from the literature showed that LGALS3BP is found in fractions of extracellular vesicles from supernatants of cells in culture and from biofluids including cerebrospinal fluid. Immunocytochemistry analysis showed that it was predominantly detected at the surface of vesicles/structures of irregular shape with a size below that of exosomes (Costa et al, PMID: 29888905; Costa et al, ref 39) and it was enriched in exomeres (Zhang et al, PMID: 29459780). In vitro assays have shown the propensity of recombinant LGALS3BP to form oligomers (Sasaki et al, PMID: 9501082).

In this manuscript, a fraction of EVs has been isolated by differential ultracentrifugation but whether LGALS3BP is present on exosomes, microvesicles, exomeres or other nanoparticles has not been shown. Therefore, it is more appropriate to replace in the manuscript "exosomes" by "extracellular vesicles fraction". The related text throughout the manuscript should also be revised by taking into consideration the information provided in the previous paragraph.

2. Figure 8b has insufficient publication quality.

The difference between the levels of endogenous LGALS3BP (SH-SY5Y cells) and the HA fusion proteins is not evident (high background and smears on the blots; the extraction buffer probably is not adequate).

Potential contaminants of EV fractions should be tested (see criteria in Lotvall et al, PMID: 25536934).

Do the mutants have a toxic effect or a loss of function?

What was the amount of EV fraction applied in the assay?

3. Line 127:

It is not obvious that the dotted staining of LGALS3BP confirms its presence in vesicles.

Immunohistochemistry would not be expected to have enough resolution to distinguish exosomes as dots. It is not possible to visualize if the dots are intracellular or extracellular. Based on the scale bar of the figure what would be the estimated size for those dots? How would it fit to the known dimensions of EVs (point 1 above)?

4. Page 12:

Concerning the secretome analysis it appears that proteins related with platelet degranulation were very enriched. Would there be some significance for this observation? On the other hand, it is puzzling that components of the endoplasmic reticulum lumen are enriched in the secretome. Would there be an explanation?

5. Figure 5:

Here, the whole extracellular medium has been tested on the CO cultures. Why was not the isolated EV fraction tested? Would it be possible to test purified LGALS3BP (from the medium and recombinant) on this system? In line 343, the conclusion "...it is influenced by the correct secretion of LGALS3BP" is probably over-interpreted.

6. Line 522:

Discussion "We also show that LGALS3BP is secreted via exosomes ...The uptake of this exosomal-secreted LGALS3BP..." is overdone in view of the results presented.

7. Section starting at line 868:

What were the culture conditions of SH-SY5Y cells? Were the cells grown in bovine serum depleted of bovine exosomes? What was the amount of exosomes produced by the cells?

Minor points:

1. Title: "Extrinsic function": the meaning is unclear.

2. In the text the figures should follow the order they appear (e.g., switch Ext Data 1b and 1a).

3. Line 573: "EOMES" instead of "TBR2" for consistency.

4. Line 784: "from" instead of "form".

5. Line 882: "cell" instead of "ell".

6. Line 887: Triton?

7. Line 888: Tween?

8. Line 891: NGS?

9. Line 922: sucrose

10. Line 1045: "Bicinchoninic".

11. Check references formatting (e.g, ref 9, 26, 33, 36, 47, etc).

12. References in the manuscript should be separated by commas.

REVIEWER COMMENTS

Reviewer #1 (Remarks to the Author):

Kyrousi et al., performs a significant amount of work in both in vitro and in vivo cortical development models to characterize the role of human LGALS3BP mutations, encoded by Galactin-3 binding protein, which interacts with members of the ECM. The role of LGALS3BP is well established in cancer biology but less is known during brain development. The main question to be addressed is understanding the mechanisms that contribute to the expanded cortical size and generation of gyri and sulci in humans. While increasing rounds of basal radial glial progenitor division has been described as a possible mechanism, this paper focuses on an extrinsic mechanism, namely the role of the extracellular matrix, in neocortical expansion and cortical folding. Therefore this paper is conceptually interesting and novel in this regard. In a nutshell, the authors provide data which support the following conclusions:

1. LGALS3BP is a human-enriched ECM molecule in neurogenic zones near NPCs shown by IHC and in situ hybridization of cortical organoids at different stages and is a secreted protein from cortical organoid vesicles.
2. iPSCs gene edited cortical organoids with patient specific LGALS3BP mutations exhibit misplacement of NPCs and neurons, including changes in in NPC divisions, radial glia markers, and other morphological changes, consistent with individuals carrying these variants with neurodevelopmental conditions showing cortical malformation.
3. LGALS3BP levels is underlying this phenotype as overexpression of LGALS3BP (both WT and mutant variants) is sufficient to misplace NPCs and neurons in mouse cortex development and slices of human fetal brain.
4. LGALS3BP secretion from exosomes is suggested to be the purported mechanism to alter the protein composition of the extracellular niche. But there is no direct evidence for this, this is the weakest part of the story, therefore I suggest this part to be removed or simplified as part of another paper.

Despite these strengths, there are also major weaknesses in the presentation of the data to fit a cohesive story. There is an overabundance of data, sometimes the organization gets unwieldy, and the results section is very long. It seems like there's two separate stories here, and it's not clear how the significant amount of 'omics data, e.g., transcriptome, proteome, secretome, come together to produce a mechanistic explanation for the cellular and phenotypic results.

We thank the reviewer for the encouraging comments. Following the reviewer's suggestion, we have included in the revised version more discussion and experiments to further support our findings. We have also edited the part with the extracellular vesicles as suggested by all reviewers. We feel that this part is now stronger and since this is the first work, to our knowledge, suggesting an EV-mediate mechanism involved in the regulation of neuronal progenitors, we thus believe that it should be included. We hope that the reviewer will share our enthusiasm for this.

Major comments:

1. To strengthen the conclusion that LGALS3BP is a human-enriched ECM molecule, the authors should provide the data, or clearly describe the previous work, that it is not expressed during mouse (or other species) cortical development. Also it is implied that the individuals with the de novo LGALS3BP mutations have higher levels of LGALS3BP, so does this mean they are gain of function mutations? What is the difference between the E370K and Y366Lfs mutations, in terms of more or less LGALS3BP secretion? Why do the authors study both; more compelling rationale in the paper should be described. The authors

should directly examine the level of LGALS3BP in the medium of Ctrl vs mutant organoids and correlate it to the expression of LGALS3BP within the organoid tissue with the mutant phenotypes.

We thank the reviewer for helping us identifying the weak points in the presentation of previous work. Following the reviewer's suggestions, to clearly state the human-enriched expression of LGALS3BP, we have modified the text in the last paragraph of the introduction and first paragraph of the results part, we have added more references of previous work and we have clearly labelled our data in Fig. 1 and Extended Data Fig.1 showing the expression of LGALS3BP at the mRNA and protein levels. Regarding the mutants of LGALS3BP, we have checked the expression levels of the ctrl and mutant LGALS3BP in COs using different approaches (sc-RNA-seq as shown in Fig.4 and Supplementary Table 1, immunohistochemistry as shown in Extended Data Fig.2, proteomics in the organoids lysates as shown in Fig.4 and Supplementary Table 2 and secretomics in lysates from the secreted proteins as shown in Fig.4 and Supplementary Table 3). Besides, following the reviewer suggestions, we performed additional analysis on the protein expression of LGALS3BP in mutant COs by calculating the number of puncta in immunofluorescence with a specific LGALS3BP antibody in CO sections at 60d (Extended Data Fig.2). All these data show the dramatic reduction of the mRNA and protein levels of the *de novo* variant of LGALS3BP (E370K) and the exon 5 KO mutant (Y366Lfs) that was generated for the purposes of this project. Also, no statistical difference was observed in the mRNA or protein levels (in the COs tissue or the secreted) of the E370K and the Y366Lfs as shown in Fig.4 and Extended Data Fig.4. Following the reviewer's suggestion and to make this point clearer, we have changed the text in the results part accordingly to emphasize these differences. We included details regarding the two different mutant iPSCs that we have generated and the reasons why we chose to generate these mutations on page 8 of the revised manuscript. We hope that the reviewer will find this information well-presented.

2. The authors examine many aspects of cellular phenotypes in cortical organoids – progenitor cell distribution, cell division, delamination, cortical thickness. But central to the story is the effect of LGALS3BP on the formation of gyri and sulci. While this is examined in the overexpression studies of the human genes in mouse, the authors do not directly examine folding in organoid culture. Perhaps this is because there is no effect on folding or organoid morphology. The authors should be explicit about this because there seems like a disconnect here. The authors should also consider adding a table to summarize the many different cellular phenotypes. This gets unwieldy very quickly.

We agree with the reviewer that central to the story is the effect of LGALS3BP on the formation of gyri and sulci and that we have not examined that in COs. This is mainly due to the limitation of this model system in accessing this kind of features – COs do not form folds since the pial surface of the neuroepithelium is not well-defined like *in vivo*. For this reason, we used the other two alternative approaches: the human cortex (MRIs from patients with LGALS3BP mutations in Fig.2) and the mouse cortex (*in vivo* overexpression of LGALS3BP in Fig. 6,7, Extended Data Fig.6). Following the reviewer's suggestion, we took special care to clearly state that in the text. We also include a summary table with the different cellular phenotypes in Extended Data Fig.8.

3. How do the authors interpret that overexpression of the WT LGALS3BP (but not mutant variant) in the mouse model does not result in major morphological differences in folding, but an effect on progenitor cell distribution? How would the mouse niche compensate for an ECM molecule it does not normally express?

We agree with the reviewer. There are no major morphological differences in the folding when the mutant variants are overexpressed. However, the same is observed also in the progenitor distribution which also doesn't differ from ctrl mice. Why the mutant LGALS3BP variants cannot recapitulate the phenotype observed when the wt form is introduced in the mouse developing cortex can be attributed to the fact that under normal conditions LGALS3BP is not expressed in the mouse and thus the

overexpression of a non-functional/less-functional variant of the gene does not result in a phenotype because it does not change the physiological expression levels. Therefore, we believe that there is no need for compensation for non-mouse ECM molecules. We have modified the text to make this part clearer and to avoid any misinterpretations from the readers. We hope that the reviewer and the editor are satisfied with our explanation.

Minor comments:

1. The format used for the references is confusing.

We agree with the reviewer that the reference style is confusing. We have chosen a “Nature” style to fit the style of the Nature group. We have now changed to the “Nature Communication” style to better fit with the rest of the papers in the journal by performing some modifications to the default setting of the style (i. e. putting brackets).

2. What do the authors mean by range and frequency in extended data figure1b?

The reviewer is right, we haven’t explained this information since these data were obtained directly from a previously published paper (Nowakowski et al 2017) using an online tool provided by the authors. In our revised manuscript we include a short description of it in the legend.

3. Figure1a shows peak of expression of LGALS3BP at 120D. However, the authors chose to perform single cell RNA seq, proteome and secretome analysis at 60D rather than 120D. The authors should make a comment to clarify this.

The reviewer is right. LGALS3BP expression is higher at 120D, this could correlate with its expression in glial cells. However, from the analysis that we have performed, we saw that a significant change in the expression levels of LGALS3BP occurs already at 40D as indicated in Fig.1a while later no significant differences were observed between the different time points (40D compared to all later time points). On the contrary, BP markers like HOPX and EOMES significantly increased after 70D. Thus we have chosen 60D for further analysis because it is a stage in COs culture where LGALS3BP is significantly increased while later events of differentiation like increase of BPs is not yet observed. We have included a sentence in the revised manuscript to clarify this point.

4. The figure legend for Figure 1c is missing.

We thank the reviewer for pointing this. We have added a description in the legend.

5. What was the sequencing depth of the single cell libraries?

We have included this information in Supplementary Table 1.

6. Page6, line#128-129, the authors write that LGALS3BP staining “confirms its presence in vesicles as recently reported”. This sentence is slightly misleading as the authors have not performed staining with markers of vesicles. Moreover, it’s not clear from the stainings whether LGALS3BP is present in the cells or around the cells?

We agree with the reviewer, this statement is misleading. Indeed we also didn’t do any co-staining with vesicle markers. However, we would like to state that the dotted staining that we observe is typical for proteins contained in vesicles and thus this expression in the developing human cortex comes in line with previous publications where they describe LGALS3BP as being present in vesicles (exosomes). We have changed this phrase in the revised manuscript to reflect our work more accurately.

7. The rationale for choosing 40d time point for overexpression studies is not very clear.

We thank the reviewer for pointing out this omission. To clarify, we have chosen the stage before the peak of expression of LGALS3BP and before the increase of the BPs (EOMES+ intermediate progenitors and HOPX+ bRGs) since we wanted to estimate whether the misplacement of BPs is due to defects in their generation or their delamination. We have included this information in the manuscript.

8. In figure 2d-f, the authors claim that there is a difference in the distribution of NPCs in the variant background and have performed immunostaining with PH3, a marker for dividing cells. However, to be sure that it's only the neural progenitor cells that are dividing, it might be better to add a marker of neural progenitor cells as well in this staining.

We agree with the reviewer that we do not show any co-staining of the PH3+ cells with a progenitor marker. This is due to technical issues since the antibodies for progenitor markers that are working nicely in the organoids and that we are using (e.g. PAX6, SOX2) are the same species as the PH3 antibody which make the co-staining not possible. We have tried additional PH3 and PAX6 commercially available antibodies but we did not succeed. However, at this stage of the CO cultures, the vast majority of PH3 mitotic cells are neural progenitors. Glial cells will be generated later as shown in many publications that performed sc-RNA-seq in cerebral organoids (Velasco et al 2019, Kanton et al 2019). To validate this in our organoid cultures we decided to check all the cells that are in M-phase and which have a specific and distinct DAPI morphology (due to chromosome concentration). All these cells are SOX2 positive in ctrl, E370K and Y366Lfs COs as shown in the following figure (Reviewer Fig.1, see arrows pointing at M-phase DAPI+ cells). Due to limited space in the Figures of the manuscript, we decided not to include these images. However, if the reviewer and editor decide that this is important to be added, we will include it. We hope that the reviewer finds this satisfying.

Reviewer Fig.1. Micrographs of sections of 60d control (a-a'), E370K (b-b') and Y366Lfs (c-c') mutant COs immunostained with SOX2, a marker of neural progenitors and DAPI, a dye that labels the DNA of the cells. Arrows depict cells that have a unique DAPI morphology because they are at the M-phase of their cycle.

9. In figure 2g-h, it is not clear what do the authors mean by apical cells normalized to apical length and basal cells normalized to apical length.

We apologized for the lack of information on this point. Since the ventricles found within the organoids are often different in size, we normalize the counting of the absolute numbers of PH3+ cells that have

been counted with a feature that will be indicative of the size of the ventricles. We, thus, decided to measure the length of the apical length of the ventricles, a feature that has been already used in previous publications (e.g. Iefremova et al 2017, Buchsbaum et al 2020). We have specified that in the materials and methods part.

10. The authors should clarify how they calculated the Z-score in their analysis.

We have included details for the z-score calculation in the materials and methods in the part “Cell and tissue quantifications”

11. Figure format/organization:

- Images in the figures are good but the organization of the panels is confusing. The authors should strictly adhere to proper citation of the figures. In some figures later panels are discussed earlier in the results. For example, figure 1b is discussed before figure 1a. Similarly, figure 3s,t are discussed before 3b-e,o and so on and so forth.

We thank the reviewer for helping us present the data in a better way. We have carefully checked this point and we made all the necessary modifications.

- Some graphs are small which makes it hard to evaluate the data (Fig.2b and c; Fig.4a and c)

We understand the point that the reviewer raises but due to limitation on the space in the figures we cannot include bigger images. However, the plot in Fig.2b and c are accompanied by quantifications and all data from Fig.4 are presented in Supplementary Tables 2 and 3. In addition, we include here a figure (Reviewer Fig.2) which shows these plots in a bigger size. We hope that the reviewer will be able to assess the quality of the plots and will understand the reasons why we decided to include only smaller plots in the figures of the paper.

Reviewer Fig.2. Plots depicted in a bigger size for evaluation of the quality of the data. (a) Plots are shown in Fig. 2b. (b) Plots are shown in Fig. 2c. (c) Plots are shown in Fig. 4a. (d) Plots are shown in Fig. 4c.

- Statistical data is not easy to see in some graphs (Fig.2l, Fig.3i, Fig.5j, Fig.6x, Fig.7w). It would be better if the asterisks signs are inserted above the bar graphs and not in between the data bars.

We understand the point of the reviewer. We have changed this according to the reviewer's suggestion in most of the graphs. However, as he/she may have noticed there is limited space in some of the figures and for this reason, we believe that there, asterisks between the data bars is the best representation. In case the reviewers and editor believe that this is an important point we will change it by increasing the number of figures, however, we feel that this will spread the data and will not help the readers.

- The sample size should be included in figure legends or figures.

We have made the appropriate modifications to the Figure legends.

Reviewer #2 (Remarks to the Author):

In this work, Kyrousi et al., examine the functional role of LGALS3BP in neocortical development using both human (organoid and patient clinical data) and mouse models. LGALS3BP was predicted to be of importance in cortical development due to the increased expression of this protein in human NPCs which is shown in this work using human organoids and primary human tissue. These authors identified genetic variants in LGALS3BP in three human individuals (E370K, E294K, E527G) and go on to show that some of this genetic variation, along with LGALS3BP overexpression, influences NPC delamination and positioning by acting extrinsically via exosome secretion. This is demonstrated using a wide variety of techniques including immunostaining, proteomics, and single-cell RNA-sequencing (scRNA-seq). Finally, these authors show that LGALS3BP overexpression in the mouse cortex impacts NPC (and progeny) position in the cortex and induces folding, while the genetic variants are less able to do so.

This is a interesting study that adds a another gene/protein to the growing list which play human-specific roles in cortical development. The authors should be commended for undertaking such a diverse set of studies to understand the role of LGALS3BP and its genetic variants. I do, however, have a number of major and minor points which should be considered/addressed.

We thank the reviewer for the kind words and constructive criticism. We have made several changes in the revised manuscript to improve our work.

Major:

1) LGALS3BP expression in mouse versus human cortex isn't shown in Extended Data Fig 1b where the callout occurs on Line 106. The authors have experience with both mouse and human systems. This should be shown by directly comparing the expression of LGALS3BP in mouse and human cortex as this is the primary rationale for this work.

We thank the reviewer for pointing out these discrepancies. We have made appropriate modifications in the revised manuscript (regarding line 106 of the previous version). We have based our hypothesis both on published data comparing the mouse vs human expression of LGALS3BP and we have also performed more experiments to support this. We include immunofluorescence in the human fetal cortex, in human iPSCs derived COs and in the mouse cortex (Fig. 1 and Extended Data Fig.1). To help the reviewers and the readers to easily understand these data we have included titles in each panel of the figures to discriminate the different model systems. We hope that now the figures are easier to follow.

2) In Figure 2A and A' there is a really obvious difference between these two patients (and thus the variants). This is not mentioned in the manuscript text. Why is the difference between these variants so large and what was the rationale for selecting E370K (the more modest phenotype) for most of the rest of the experiments? Moreover, E294K (the more significant phenotype) has more modest effects in the mouse (Figure 7n, for example) compared to E370K. These points should, at minimum, be addressed with further discussion.

We agree with the reviewer that the two patients have differences in the severity of their phenotype according to their MRI morphometric analysis. This may be because individual 2 is very young. This patient was only one year old when imaged which, due to incomplete myelination, makes it difficult to properly assess the results thus the MRI analysis may not be as accurate. This phenotype could have been clearer in older stages of development. Unfortunately, MRIs of this patient at an older age were not available. For this reason, we decided to choose the *de novo* mutation found in individual 1 for our further analysis. We also decided to compare these data with an exon 5 KO variant of the LGALS3BP (Y366Lfs) which would be more representative for all the variants found in exon 5 of the gene. We have tested the mRNA and protein levels of the two mutant lines and we show the reduction of LGALS3BP expression suggesting that both the point mutation and the exon 5 KO result in loss of function. Besides, most of our results show no statistical differences between the E370K and Y366Lfs iPSC lines which suggests that the phenotype is due to loss of LGALS3BP expression due to changes in the exon 5 of the gene and not due to the certain point mutation. For the above reasons, we decided to include the E294K mutant only in the *in vivo* analysis which is easier and faster.

According to our data, overexpression of the two mutant LGALS3BP in the developing mouse brain cannot fully recapitulate the phenotype of the wt form. Indeed both E370K and E294K show no statistical differences in the number of mislocated NPCs compared to control (Fig.7e) and much less cortical folding (Fig.7n). We believe that this is due to the fact that LGALS3BP have a human-specific function in the delamination of NPCs. Thus, in a system where LGALS3BP is normally not expressed (Fig.1, several publications cited in the manuscript) the overexpression of the mutant forms, that according to our previous analysis (Fig. 4,5) are dominant-negative/less-functional, should not have a function *in vivo* i. e. mouse NPCs that do not normally delaminate they still keep their normal position and folds are not generated in a normally smooth cortex. So the comparison of the phenotype between the 2 mutant LGALS3BP forms should not differ and indeed no statistical differences are observed only minor changes which can be also explained by technical differences.

We hope that the reviewer will understand and accept our rationale. Finally, following the reviewer's suggestion, we have included the above information and we made changes throughout the text to support our experimental approach, results and conclusions.

3) In Extended Figure 2E LGALS3BP doesn't appear to be reduced in the E370K samples while Y366Lfs seems to be much lower. Quantitative data should be provided to support the claim that both E370K and Y366Lfs samples have reduced LGALS3BP. Moreover, in Figure 4A, in the case of E370K LGALS3BP is not significantly different in the proteome data. This is consistent with the E370K data in Extended Figure 2E. Are all of these variants loss-of-function? Overall, the impacts of the different mutations need to be clarified/discussed in more detail.

The mutant lines that were generated for the purposes of the project were loss of function mutations since the expression levels of LGALS3BP were reduced as shown from our analysis. Specifically, we have checked the expression levels of the ctrl and mutant LGALS3BP in COs using different approaches (sc-RNA-seq as shown in Fig.4 and Supplementary Table 1, immunohistochemistry as shown in Extended Data Fig.2, proteomics in the organoids lysates as shown in Fig.4 and Supplementary Table 2 and secretomics in lysates from the secreted proteins as shown in Fig.4 and Supplementary Table 3). Besides, following the reviewer suggestions we performed additional analysis on the protein expression

of LGALS3BP in mutant COs by calculating the number of puncta in immunofluorescence with a specific LGALS3BP antibody in CO sections at 60d and we included these data in Extended Data Fig.2. We show now the statistical difference in the expression of LGALS3BP between the control and mutant COs. These data also show that the E370K (a heterozygous iPSC line) has a smaller reduction in the expression levels comparing to Y366Lfs (a homozygous iPSC line). This is in line with the proteome data in Fig.4. The reviewer correctly notices that the E370K is border line in the significant difference in the proteome. We believe that this is because proteomic analysis is a very sensitive technique. We have also applied stringent borders on what we accept as significant. In light of these new data and the reviewer's suggestions, we made appropriate changes in the text to provide clarifications on the impact of the different mutations.

4) In a number of cases in the organoid data, the images shown either seem 1) not to be similar to similar comparisons; or 2) the representative data shown doesn't seem to well-represent the quantitative data shown; or 3) it is unclear what is being shown in the figures. For example, A) in Fig 2i'',j'',k'' impaired delamination is being described but it's not clear what is being shown in these panels; B) in Fig 3 c/d/c' these sections don't look like similar comparisons and it's unclear what is being shown here; why are their nuclei below the apical border?; and C) Extended Data Fig 3b-e'/f-i'/j-m' the images shown don't seem to represent the quantitative data and are not all like comparisons.

We thank the reviewer for helping us presenting our data. A) Following the suggestions we have made appropriate modifications in Fig. 2 legend. B) Fig.3c/d/c' represent the same picture where LGALS3BP was overexpression in ctrl COs and the sections were stained with phalloidin (in c only channels red and blue are depicted, in d channels red, green and blue, in c' magnification from c as shown with the white box). We have changed the figure legend to make this point clearer and we hope that the reviewer is now satisfied. In some cases, cells were submerged to the ventricular space as the reviewer noticed, possibly due to the loss of a properly formed apical membrane. We have mentioned that in the manuscript. C) We agree with the reviewer that some of the pictures shown in these panels have differences in the number of GFP+ electroporated cells. This is due to technical reasons since each of the ventricles within the same organoid is electroporated independently and thus it is hard to regulate the amount of DNA injected which results in differences in the electroporation size. Besides, the size of each ventricle varies which also affects the amount of DNA injected. However, we have normalized the quantifications to the size of the electroporation to avoid problems in the comparisons and we have included many organoids from different batches and many different ventricles from each organoid. We specify these in the legend of the revised manuscript. We also use these data as supportive to the data obtained from the different model systems that were used in this study and for this reason we included these quantifications in the supplementary information and not in the main Figures of the manuscript. We hope that the reviewer will understand the difficulties of this experimental approach.

5) Loss of LGALS3BP causes centriole accumulation in cancer cells (Fogeron, Nat Comm, 2013) and centrioles are important for organizing cilia. In Fig. 3v-x ARL13B (a known centriole/cilia marker) seems to be increased in the mutant organoids. Cilia dysregulation is known to impact cortical development and cilia are important for responses to extrinsic cues. The authors should investigate whether the impacts of LGALS3BP mutations are due to dysregulated cilia.

We agree with the reviewer that according to Fogeron et al 2016 LGALS3BP may have a role in cilia function and that according to our data it seems that cilia are increased. We followed the reviewer's suggestion and we have investigated this further. We have calculated the number of cilia in control and mutant COs and indeed we validate that there is a statistical difference in the total number of cilia in the E370K and Y366Lfs COs, we have also observed a reduction in their length. These data, included in the revised manuscript can be explained by the increased number of progenitors found in the VZ of the

mutant COs as described in our manuscript. However, we did not observe any dysregulation of cilia like disorientation. Besides, analyzing the sc-RNA-seq, proteomic and secretomic data we did not observe any cilia-related dysregulated genes or proteins (no GO terms related to cilia pops-up in our analysis – Fig.4, Extended Data Fig.4 and Supplementary Tables 1,2,3). From all the above we believe that cilia most likely are not causative of the phenotype, in contrast, the small differences observed may be a secondary effect. We included this analysis in the revised manuscript and we discuss this in our manuscript.

6)Folding is shown to only occur in 62% of the mouse brains at 13 dpe using LGALS3BP overexpression. Why is there so much variability in the folding phenotype? Is it due to the number of electroporated cells per brain or the degree of overexpression or something else? This should be investigated and clarified.

The reviewer correctly notices that in Fig.6x there is a 62% folding which includes both 6dpe and 13dpe stages (and not only 13dpe), while almost 80% when analysing the 6dpe alone (Fig.7n). The difference mainly reflects the minor folds (the percentage of the major folds remain the same). A possible explanation may be that some minor folds may not be maintained in later developmental stages given that i) the overexpression is transient and ii) the mouse cortex normally is smooth and thus intrinsic mouse mechanisms may take over and mask this phenotype at later developmental stages. Following the reviewer's suggestion, we discuss this point in the revised manuscript.

Minor:

1)In Fig1c-e, the in situ data is so low magnification its not very clear what is being shown. Higher magnification images would be beneficial.

We have included higher magnifications of these data in Extended Data Fig.1 (panels d-e).

2)In Figure 5 there are no images of Ctrl COs with Ctrl medium and thus difficult to interpret these data. **We agree with the reviewer that in Fig.5 we have only included images for the E370K+ctrl medium and Y366Lfs+ctrl medium and no Ctrl COs with Ctrl medium. However, these images are included in other parts of the manuscript (Fig.2d-f for PH3 staining, Fig.3j-l' for phalloidin and Extended Data fig. 2z-ab' for NEUN). Let us point out that this analysis has been done in many different batches and COs and has been quantified (Fig. 2g,h, Extended Data Fig. 2ac and Fig. 3y respectively). The reviewer can appreciate from these imagings/quantifications that no big differences have been observed between different batches. Thus, considering our limited space, we believe that these images would be unnecessary. In case that the reviewers and editor disagree we can add these panels.**

3)In the scRNA-seq data in Figure 4, What is the contribution of individual organoids or were organoids combined? How robust are these data across different organoids? This should be clarified. Also, in Extended Data Fig 4 it's not clear how bRG are being defined and how signature scores are being computed (I don't see this in the methods).

For addressing this point we have included new data in Supplementary Table 1 and two new plots in the Extended Data Fig.4 where we show the contribution of each CO in the cell identity, gene numbers and transcript numbers. Besides, we have included information regarding the bRG signature calculation in the materials and Methods section.

4)The quantitative data in Figure 7n has no statistical testing and 7w has low N when using proportions; one more or less animals in each group could dramatically change these data.

We thank the reviewer for helping us improving our statistical analysis. Following the reviewer's suggestion, we have included the statistical analysis in Fig.7n and we have included more n in Fig7w.

5) Finally, in the HA-LGALS3BP blot in Figure 8b there is a blob where the LGALS3BP band would be. That is unfortunate, because it's not possible to tell if there is a band there or not. This is important because HA-LGALS3BP has the largest effect in functional studies. Ideally, this should be replaced/repeated.

We agree with the reviewer about the quality of Figure 8b. Following his/her suggestion we have repeated the western blot in extracellular vesicles extracts using a specific antibody against HA to replace the previous plots that had a high background. We present in our revised manuscript a clearer blot where it is shown that all extracts except HA alone overexpression immunoreact with HA antibody (band at the size of the LGALS3BP protein), verifying that the overexpressed proteins are produced and secreted in the extracellular vesicles. Unfortunately, the LGALS3BP antibody does not work very well in western blots (a previously used lot of LGALS3BP antibody is not available anymore) and thus we could not repeat that western blot. However, we include a source data Figure which includes all the uncropped blots where it is clear that there is no blob in the blot with the LGALS3BP antibody. We hope that the reviewer will understand this technical difficulty.

Reviewer #3 (Remarks to the Author):

In this manuscript the authors have investigated the functional role of LGALS3BP in brain development. Evidence showed that LGALS3BP is enriched in human neural progenitor cells, LGALS3BP mutations found in neurodevelopmental diseases were associated with cortical malformations and LGALS3BP controlled apical anchoring of human progenitor cells. Furthermore, a set of evidence (comparing wt vs LGALS3BP mutants) including the molecular signatures of the secretome from cerebral organoids (CO), the effect of CO medium on CO differentiation and the effect of exosome-enriched fractions (from SH-SY5Y cells) on organotypic mouse brain slices led the authors to propose that LGALS3BP would be secreted via exosomes, relevant for its biological function.

An impressive variety of complementary techniques and biological systems (humans, human-specific and mouse model) have been studied. In the literature, LGALS3BP has been more widely studied in cancer and only more recently it started to be investigated in the nervous system and neurological diseases. In view of this, results are very innovative and constitute a very significant advancement to elucidate the role of LGALS3BP in human corticogenesis.

The work is described with great detail and the results are in general of high quality and convincing. However, there are the following concerns:

We thank the reviewer for helping us improving our manuscript. We have performed new experiments and we have included more discussion in the revised manuscript according to reviewers' suggestions.

1. Extracellular vesicles are released from cells and are heterogeneous (Cocozza et al PMID: 32649878); they include exosomes (endosomal origin; 50-150 nm diameter), microvesicles (budding of plasma membrane) and, more recently, exomeres (30-50 nm diameter) have been identified. Data from the literature showed that LGALS3BP is found in fractions of extracellular vesicles from supernatants of cells in culture and from biofluids including cerebrospinal fluid. Immunocytochemistry analysis showed that it was predominantly detected at the surface of vesicles/structures of irregular shape with a size below that of exosomes (Costa et al, PMID: 29888905; Costa et al, ref 39) and it was enriched in exomeres (Zhang et al, PMID: 29459780). In vitro assays have shown the propensity of recombinant LGALS3BP to form oligomers (Sasaki et al, PMID: 9501082).

In this manuscript, a fraction of EVs has been isolated by differential ultracentrifugation but whether LGALS3BP is present on exosomes, microvesicles, exomeres or other nanoparticles has not been shown. Therefore, it is more appropriate to replace in the manuscript "exosomes" by "extracellular vesicles

fraction". The related text throughout the manuscript should also be revised by taking into consideration the information provided in the previous paragraph.

We thank the reviewer for helping us in characterizing the extracellular vesicles part of our work. We agree with the reviewer that this work is not focused specifically on exosomes but rather on secreted vesicles. Thus, following this suggestion, we have replaced throughout all manuscript the term "exosomes" with "extracellular vesicles (EVs)".

2. Figure 8b has insufficient publication quality.

The difference between the levels of endogenous LGALS3BP (SH-SY5Y cells) and the HA fusion proteins is not evident (high background and smears on the blots; the extraction buffer probably is not adequate).

Potential contaminants of EV fractions should be tested (see criteria in Lotvall et al, PMID: 25536934).

Do the mutants have a toxic effect or a loss of function?

What was the amount of EV fraction applied in the assay?

We agree with the reviewer that Figure 8b in the previous version was not of very good quality. Following his/her suggestion, we have repeated the western blot in extracellular vesicles extracts using a specific antibody against HA to replace the previous plots that had a high background. We present in our revised manuscript a clearer blot where it is shown that all extracts except HA overexpression immunoreact with HA antibody (band at the size of the LGALS3BP protein), verifying that the overexpressed proteins are produced and secreted in the extracellular vesicles. Besides, we include a source data Figure which includes all the uncropped blots where it is clear that our new data are now of high quality without background or unspecific bands. For quality check of the extracellular vesicle preparation, we have performed a detailed analysis of the number and size of the vesicles isolated via nanoparticle tracking analysis using a Particle Metrix ZetaView® (Particle Metrix GmbH, Inning am Ammersee, Germany) equipped with a 520 nm laser. We show that the extracellular vesicles that we purify with the method described in the paper are around 150nm – typical for exosomes and small extracellular vesicles as the reviewer mentions in the first comment. We show these data in a new figure (Extended Data Figure 8). Additional details are included in the materials and methods section, for example the number of cells producing the vesicles and the quantification of vesicles used for treatment in different conditions (2 million cells cultures in 15ml culture media). We do not observe any toxic effect of the mutant forms – no significant cell death in the SH-SY5Y cell cultures. For western blot experiments we always use the 1/10th of the EVs produced and for the treatment on the organotypic cultures, we always use all the amount of EVs produced per batch.

3. Line 127:

It is not obvious that the dotted staining of LGALS3BP confirms its presence in vesicles. Immunohistochemistry would not be expected to have enough resolution to distinguish exosomes as dots. It is not possible to visualize if the dots are intracellular or extracellular. Based on the scale bar of the figure what would be the estimated size for those dots? How would it fit to the known dimensions of EVs (point 1 above)?

We agree with the reviewer that the statement in line 127 is misleading. Thus we have modified the manuscript accordingly. Following reviewers' suggestion, we have measured the size of the dots which are between 100nm and 300nm. However, as the reviewer already mentioned, immunohistochemistry is not the ideal method for this analysis since it is very hard to distinguish whether these dots are one single vesicle or more very close to each other. We have however analyzed the vesicles isolated from SH-SY5Y cells which are on average at the size of 100nm which fit the size of the exosomes and small extracellular vesicles as the reviewer correctly states in his/her first comment. We also show by western blot that LGALS3BP is found in these vesicles. These new data are included in Fig.8 and Extended Data Fig.8

4. Page 12:

Concerning the secretome analysis it appears that proteins related with platelet degranulation were very enriched. Would there be some significance for this observation? On the other hand, it is puzzling that components of the endoplasmic reticulum lumen are enriched in the secretome. Would there be an explanation?

We agree with the reviewer that these data are puzzling. However, we believe that these can be artifacts or limitations of the proteomic/secretomic analysis in COs. In fact, we have used tools like PANTHER for GO terms analysis which is a well-established method for proteomic and secretomic analysis which, however, has been designed to compute data from *in vivo* models while not enough data from CO have yet been generated to correct for these limitations. Thus, we believe that platelet degranulation may not be a significant finding given that COs do not have blood cells with the protocol used in this study. Regarding ER lumen, this can also be an artifact due to the experimental approach. Specifically, the COs' medium for secretomic analysis was collected 4 days after the last change of medium to enrich for secreted proteins, however, the medium may be contaminated from endoplasmic proteins that are originated from disruption of dead cells which are present in CO cultures as described by almost all publications using COs (one of the big limitations of the system is the lack of vascular system which helps the survival of the cells). On the contrary, our interpretation of these results that the phenotype is mediated by extracellular vesicles are very abundant in the data (i. e. enrichment of extracellular space, extracellular matrix, extracellular vesicles) with the highest FDR and hold change enrichments. We feel that emphasis on the possibly false-positive data about platelets and ER may confuse the readers and thus, we decided not to discuss further these in the manuscript. However, in case the reviewers and editor believe that this would be a substantial addition to the paper we will include it.

5. Figure 5:

Here, the whole extracellular medium has been tested on the CO cultures. Why was not the isolated EV fraction tested? Would it be possible to test purified LGALS3BP (from the medium and recombinant) on this system? In line 343, the conclusion "...it is influenced by the correct secretion of LGALS3BP" is probably over-interpreted.

We agree with the reviewer. We did not specifically attempt to rescue the phenotype of the mutant COs by giving the recombinant protein or the extracellular vesicles but rather we performed that by administering the whole extracellular medium. This was done mainly due to technical reasons, namely the very long time in culture. In particular, we sought to administer the wt LGALS3BP to the mutant COs for the whole period of the CO development from the early time points of the neuronal differentiation (16d) to the time point when the whole previous analysis in Fig.2, 3, 4 was done (62d). Also, to avoid masking the phenotype by degradation of the administrated wt protein and the accumulation of the mutant protein which was still produced from the mutant COs we were changing the medium every day from 16-62d. Moreover, it would be technically very challenging to isolate and purify every day exosomes/EVs for all this period (46d in total). We hope that the reviewer understands this difficulty. Regarding the overstatement that we had in the previous manuscript, we have now made the appropriate modifications to the text to more accurately describe our results. We thank the reviewer for helping us in this direction.

6. Line 522:

Discussion "We also show that LGALS3BP is secreted via exosomes ...The uptake of this exosomal-secreted LGALS3BP..." is overdone in view of the results presented.

We thank the reviewer for helping us presenting our data more clearly. Following the reviewer's suggestion, we have modified this sentence: "We also show that mutations in LGALS3BP change the composition of secreted proteins, that it is present in extracellular vesicles and that treatment with extracellular vesicles expressing LGALS3BP is sufficient to reproduce the phenotype observed upon manipulation of its expression in vivo"

7. Section starting at line 868:

What were the culture conditions of SH-SY5Y cells? Were the cells grown in bovine serum depleted of bovine exosomes? What was the amount of exosomes produced by the cells?

Following the reviewer's suggestion, we include details on the culture conditions of SH-SY5Y cells in the materials and methods. Yes, we do use FBS depleted of bovine exosomes, the full protocol is now included in the revised manuscript. Regarding the number of exosomes that were produced from the cells, we performed a nanoparticle tracking analysis (NTA), using a Particle Metrix ZetaView® (Particle Metrix GmbH, Inning am Ammersee, Germany) equipped with a 520 nm laser. We include details in the materials and methods and data in a new Figure that we include in the revised manuscript (Extended Data Figure 8).

Minor points:

1. Title: "Extrinsic function": the meaning is unclear.

We have chosen a new title following the reviewer's suggestion: "The extracellular function of LGALS3BP in neural progenitors during brain development"

2. In the text the figures should follow the order they appear (e.g., switch Ext Data 1b and 1a).

We thank the reviewer for pointing this out. We made all the appropriate modifications to the figures.

3. Line 573: "EOMES" instead of "TBR2" for consistency.

We changed this

4. Line 784: "from" instead of "form".

We have changed this typo

5. Line 882: "cell" instead of "ell".

We have changed this typo

6. Line 887: Triton?

We have included info for this reagent

7. Line 888: Tween?

We have included info for this reagent

8. Line 891: NGS?

We have included info for this reagent

9. Line 922: sucrose

We have changed this typo

10. Line 1045: "Bicinchoninic".

We have changed this typo

11. Check references formatting (e.g, ref 9, 26, 33, 36, 47, etc).

We agree with the reviewer that the reference style is confusing. We have chosen a “Nature” style to fit the style of the Nature group. We have now changed to the “Nature Communication” style to better fit with the rest of the papers in the journal by performing some modifications to the default setting of the style (i. e. putting brackets).

12. References in the manuscript should be separated by commas.

Please see the answer in the previous comment

REVIEWER COMMENTS

Reviewer #1 (Remarks to the Author):

The study by Kyrousi and colleagues provides in-depth cellular and molecular mechanistic insight regarding the role of LGALS3BP variants in cortical development, gyrification and delamination. The use of multiple models from organoids to mouse to man is compelling. The revised experiments, textual clarification, and additional details in the methods and figures strengthen the conclusions of the manuscript. I have no further concerns.

Reviewer #2 (Remarks to the Author):

This reviewer is satisfied that their concerns have been suitably addressed. However, one minor point. Figure 1H was changed in this version of the manuscript and now the arrows don't seem to point to positive cells. The authors should confirm this is the intended placement of these arrows.

Reviewer #3 (Remarks to the Author):

The authors have addressed all points adequately and have added new relevant data to the manuscript. In point 4 the reviewer agrees with the authors that it is not necessary to include the discussion in the manuscript. Overall, the manuscript presents extensive new and significant novel data that unveil the importance of LGALS3BP in brain development. The work is of high quality and is considered to be suitable for publication.

Reviewer #4 (Remarks to the Author):

Kyrousi and coworkers investigate the role of Galectin-3 Binding ProteinL (GALS3BP), and how individuals with LGALS3BP de novo variants exhibit altered cortical morphology. They report that LGALS3BP regulates the position of NPCs, and show that LGALS3BP-related mechanisms involve the extracellular matrix. This is a very interesting study, novel approach, state of the art methodology, while some issues should be addressed:

1) The study includes two individuals with LGALS3BP variants. The overlap in brain structure variation is not evident. This could be more adequately discussed.

2) The authors use Gyrification Index (LGI) cluster analysis which reveals changes in LGI in two participants. However, the LGI measure is suboptimal. What is the rationale for the current imaging analytical protocol? LGI is more difficult to align to the biological phenomenon of cortical folding. It would be interesting to apply the sulcal depth measure from FreeSurfer. Sulcal depth captures sulcal morphology, reflecting the convexity or concavity of any given point on the cortical surface, as captured with FreeSurfer cortical processing stream (Fischl 2002). This measure is driven by primary gyral and sulcal patterns and is independent on the accurate identification of grey and white matter borders, and captures complex folding patterns of the cerebral surface. LGI is difficult to interpret.

3) The human imaging phenotypes focus on cortical thickness, while cortical surface area is not investigated. This is difficult to understand, since there are different genetic determinants of cortical area versus cortical thickness.

4) The study reports findings based on iPSC experiments, while several of the claims are based on human cortex. The limited translation to the development of the human cortex involving complex regulation of millions of neurons should be mentioned. In particular, how are the main outcome variables (Cortical thickness and gyrification) from the human carriers reflected in the phenotypes

assessed in iPSC neurons? There seems to be a lack of link between human and experimental outcome measures which should be acknowledged. The potential implications for schizophrenia and autism spectrum disorders could also be discussed in more detail, in line with more recent findings (now referencing papers from 2014).

5) They also apply a mouse model, which may be relevant to study cortical thickness, but not gyrification? In the introduction. The authors highlight the specific brain characteristics of primates ("one possible explanation for the massive increase in cortical size in primates was proposed to be..."). And in results, they state that mouse cortex has a limited amount of BPs, is not as expanded as the human cortex and lacks folds. Still, they are using mouse models. There may be molecular or sequence overlap, but the neuronal phenotype overlap is lacking, making this section of the study of limited impact. This should be better clarified as a limitation.

Rebuttal letter

Reviewer #1 (Remarks to the Author):

The study by Kyrousi and colleagues provides in-depth cellular and molecular mechanistic insight regarding the role of LGALS3BP variants in cortical development, gyrification and delamination. The use of multiple models from organoids to mouse to man is compelling. The revised experiments, textual clarification, and additional details in the methods and figures strengthen the conclusions of the manuscript. I have no further concerns.

We thank the reviewer for the kind comments.

Reviewer #2 (Remarks to the Author):

This reviewer is satisfied that their concerns have been suitably addressed. However, one minor point. Figure 1H was changed in this version of the manuscript and now the arrows don't seem to point to positive cells. The authors should confirm this is the intended placement of these arrows.
We thank the reviewer for noticing this mistake. We made the appropriate changes in the revised figure.

Reviewer #3 (Remarks to the Author):

The authors have addressed all points adequately and have added new relevant data to the manuscript. In point 4 the reviewer agrees with the authors that it is not necessary to include the discussion in the manuscript. Overall, the manuscript presents extensive new and significant novel data that unveil the importance of LGALS3BP in brain development. The work is of high quality and is considered to be suitable for publication.

We thank the reviewer for the kind comments.

Reviewer #4 (Remarks to the Author):

Kyrousi and coworkers investigate the role of Galectin-3 Binding ProteinL (GALS3BP), and how individuals with LGALS3BP de novo variants exhibit altered cortical morphology. They report that LGALS3BP regulates the position of NPCs, and show that LGALS3BP-related mechanisms involve the extracellular matrix. This is a very interesting study, novel approach, state of the art methodology, while some issues should be addressed:

1) The study includes two individuals with LGALS3BP variants. The overlap in brain structure variation is not evident. This could be more adequately discussed.

We agree that the is not evident overlap in the brain structure of the two patients, however, in both cases, we observe the same phenotype i.e. changes in cortical thickness, gyrification index, surface area, and sulcal depth. These data come from our previous and new analysis (see also point 2 and 3). We understand the criticism raised by the reviewer and we follow his suggestion on discussing in more detail these findings in our revised manuscript.

2) The authors use Gyrification Index (LGI) cluster analysis which reveals changes in LGI in two participants. However, the LGI measure is suboptimal. What is the rationale for the current imaging analytical protocol? LGI is more difficult to align to the biological phenomenon of cortical folding. It would be interesting to apply the sulcal depth measure from FreeSurfer. Sulcal depth captures sulcal morphology, reflecting the convexity or concavity of any given point on the cortical surface, as captured with FreeSurfer cortical processing stream (Fischl 2002). This measure is driven by primary

gyral and sulcal patterns and is independent on the accurate identification of grey and white matter borders, and captures complex folding patterns of the cerebral surface. LGI is difficult to interpret.

We thank the reviewer for this interesting comment. We used the Freesurfer pipeline coupled with the local GI measure because it describes the abnormal patterns of the cortical mantle, as we had previously observed in polymicrogyria (doi: 10.1093/cercor/bhx036) and in a group-level analysis on PCDH19-mutated patients (doi:10.1093/cercor/bhaa177). According to the Reviewer's suggestion, we have now performed the analysis of sulcal depth, through which we found additional interesting findings in the cortical mantle in both patients, as described in Figure 2 and the revised manuscript.

3) The human imaging phenotypes focus on cortical thickness, while cortical surface area is not investigated. This is difficult to understand, since there are different genetic determinants of cortical area versus cortical thickness.

We agree with the reviewer on this point. As studies by Panizzon and team have suggested, we have now completed the analysis on the cortical surface area and added the results in Figure 2 and the revised manuscript.

4) The study reports findings based on iPSC experiments, while several of the claims are based on human cortex. The limited translation to the development of the human cortex involving complex regulation of millions of neurons should be mentioned. In particular, how are the main outcome variables (Cortical thickness and gyrification) from the human carriers reflected in the phenotypes assessed in iPSC neurons? There seems to be a lack of link between human and experimental outcome measures which should be acknowledged. The potential implications for schizophrenia and autism spectrum disorders could also be discussed in more detail, in line with more recent findings (now referencing papers from 2014).

We thank the reviewer for helping us in improving our manuscript. Indeed, some of the findings in humans seem not to be very similar to the experimental outcome. Our previous and new analysis on the gyrification, thickness, surface area and sulci depth that we include in the revised manuscript indicate a specific defect in cortical formation in patients with LGALS3BP variants. Given that modeling of cortical folding in organoids is not possible since they lack folds and that in mice is difficult to translate the exact human-specific phenotype, we are using these models to focus mainly on the number and distribution of the neural progenitor cells and the neuronal migration and we rely on the human data analysis for the rest in an effort to use the advantages of each model system on our benefit and to try to speculate on the actual mechanisms that are involved in human corticogenesis and in brain malformations. We hope that the reviewer will understand the limitation that we have faced and will agree with our approach. Following the reviewer's suggestion, we made this point clearer in the revised manuscript. We have also included more recent references in the discussion for the potential implication of LGALS3BP on neurodevelopment and psychiatric disorders as the reviewer suggested.

5) They also apply a mouse model, which may be relevant to study cortical thickness, but not gyrification? In the introduction. The authors highlight the specific brain characteristics of primates ("one possible explanation for the massive increase in cortical size in primates was proposed to be..."). And in results, they state that mouse cortex has a limited amount of BPs, is not as expanded as the human cortex and lacks folds. Still, they are using mouse models. There may be molecular or sequence overlap, but the neuronal phenotype overlap is lacking, making this section of the study of limited impact. This should be better clarified as a limitation.

We agree with the reviewer that mice lack folds and their cortex is more restricted and with a limited number of neurons comparing to the gyrified cortex of other primates and especially humans. The reason why we are using the mouse model in this context is that we believe that is essential to study the *in vivo* function of the genes and not to rely completely on *in vitro* studies.

Given that there is a limitation in the use of the human as a model due to ethical issues and the difficulties of the use of other primates we believe that the mouse is a good model to use. Besides, many scientists have used the mouse model to study human cortical development including neuroprogenitor number, neuronal migration, and gyrification (Stahl et al 2013; Florio et al 2015; Suzuki et al 2018; Vaid et al 2018; and others). In addition, we believe that LGALS3BP main function is on the neural progenitor numbers and distribution and that changes in these parameters may be essential for the evolutionary development of the human cortex. However, following the reviewer's suggestions we have modified the manuscript to discuss these issues and to avoid any misinterpretations from the readers. We hope that the reviewer will agree with our decision.

REVIEWERS' COMMENTS

Reviewer #4 (Remarks to the Author):

The authors have addressed all my comments adequately. They have added new relevant data and revised the text to the manuscript.